# Evaluation of a Low-Cost Dryer for a Low-Cost Optical Particle Counter

Miriam Chacón-Mateos[1], Bernd Laquai[1], Ulrich Vogt[1], and Cosima Stubenrauch[2]

[1]Department of Flue Gas Cleaning and Air Quality Control, University of Stuttgart, Stuttgart, 70569, Germany
[2]Institute of Physical Chemistry, University of Stuttgart, Stuttgart, 70569, Germany

*Correspondence to*: Miriam Chacón-Mateos (miriam.chacon-mateos@ifk.uni-stuttgart.de)

**Abstract.** The use of low-cost sensors for air quality measurements has become very popular in the last decades. Due to the detrimental effects of particulate matter (PM) on human health, PM sensors like photometers and optical particle counters (OPC) are widespread and have been widely investigated. The negative effects of high relative humidity (RH) and fog events
in the mass concentration readings of these types of sensors are well documented. In the literature, different solutions to these problems - like correction models based on the Köhler theory or machine learning algorithms - have been applied. In this work, an air pre-conditioning method based on a low-cost, thermal dryer for a low-cost OPC is presented. This study was done in two parts. The first part of the study was conducted in the laboratory to test the low-cost dryer under two different scenarios. In one scenario, the drying efficiency of the low-cost dryer was investigated in the presence of fog. In the second scenario,
experiments with hygroscopic aerosols were done to determine to which extent the low-cost dryer reverts the growth of hygroscopic particles. In the second part of the study, the PM10 and PM2.5 mass concentrations of an OPC with dryer were compared to gravimetric measurements and a continuous Federal Equivalent Method (FEM) instrument in the field. The feasibility of using univariate linear regression (ULR) to correct the PM data of an OPC with dryer during field measurement was also evaluated. Finally, comparison measurements between an OPC with dryer, an OPC without dryer, and a FEM
instrument during a real fog event are also presented. The laboratory results show that the sensor with the low-cost dryer at its inlet measured an average of 64 % and 59 % less PM2.5 concentration compared to a sensor without the low-cost dryer during the experiments with fog and with hygroscopic particles, respectively. The outcomes of the PM2.5 concentrations of the low-cost sensor with dryer in laboratory conditions reveal, however, an excess of heating compared to the FEM instrument. This excess of heating is also demonstrated in a more in-depth study on the temperature profile inside the dryer. The correction of
the PM10 concentrations of the sensor with dryer during field measurements by using ULR showed a reduction of the maximum absolute error (MAE) from 4.3 µg m$^{-3}$ (raw data) to 2.4 µg m$^{-3}$ (after correction). The results for PM2.5 make evident an increase in the MAE after correction: from 1.9 µg m$^{-3}$ in the raw data to 3.2 µg m$^{-3}$. In light of these results, a low-cost, thermal dryer could be a cost-effective add-on that could revert the effect of the hygroscopic growth and the fog in the PM readings. However, special care is needed when designing a low-cost dryer for a PM sensor to produce FEM similar PM
readings, as high temperatures may irreversibly change the sampled air by evaporating the most volatile particulate species and thus deliver underestimated PM readings. New versions of a low-cost dryer aiming at FEM measurements should focus

on maintaining the RH at the sensor inlet at 50 % and avoid reaching temperatures higher than 40 °C in the drying system. Finally, we believe that low-cost dryers have a very promising future for the application of sensors in citizen science, sensor networks for supplemental monitoring, and epidemiological studies.

**1 Introduction**

The use of particulate matter (PM) sensors has increased significantly in the last decade. They are widely applied in citizen science projects (Lukeville, 2019; Schaefer et al., 2020), as part of sensor networks (English et al., 2020; Gulia et al., 2020), and also for educational purposes in schools and universities to raise awareness about air quality in the young generations (Castell et al., 2021; Höfner and Schütze, 2021). Moreover, new fields of application are emerging as sensors achieve better

performances thanks to new sensor developments and new methods for data post-processing. Researchers are currently investigating the use of low-cost sensors for smart city management (Toma et al., 2019), supplemental monitoring for official measurement stations (Castell et al., 2017; Liu et al., 2019), and personal exposure (Steinle et al., 2015; Novak et al., 2020). The accuracy needed for certain applications e.g. regulatory air quality monitoring or environmental epidemiology is at this moment one of the limiting factors for the use of low-cost sensors.

The most widely used measurement principle of PM low-cost sensors is light scattering, and the most common type of low-cost sensors used in air quality research are photometers (usually nephelometers) and optical particle counters (OPCs). Photometers measure relative concentrations by detecting the combined light scattered from many particles at once (Hinds, 1999). In nephelometers, particles pass through a sensing volume as a group of particles, and the particle concentration is determined by the intensity of the total scattered light registered by the photodetector. On the contrary, in OPCs individual

particles generate a pulse on the photodetector. The number of pulses is proportional to the number of particles per unit volume and the intensity of the pulses to the size of the particles (Li, 2019). The accuracy of outdoor air measurements with light scattering instruments is seriously influenced by the relative humidity (RH) due to the water uptake of hygroscopic aerosols, and due to fog events (Jayaratne et al., 2018).

Fog is defined as visible aerosols consisting of tiny water droplets or ice crystals in the order of micrometres suspended in air

(Spiridonov and Ćurić, 2021). During fog events, the air is saturated with water vapor and the RH is around 100 %. Water droplets can substantially falsify the number and the size of the particles detected with light scattering instruments. An example can be seen in Fig. 1a, where the one-minute average PM concentration registered by a light scattering aerosol spectrometer, model 1.108 from the company GRIMM GmbH (Germany), during a fog event is presented. As can be seen, mass concentrations are extremely high, especially the PM10 values, which reach magnitudes of $10^4$-$10^5$ µg m⁻³. PM2.5 and PM1

are in the range of $10^2 - 10^3$ and $10 - 10^2$ µg m⁻³, respectively. In Fig. 1b it is shown that most of the detected particles during that fog event were smaller than 1 µm. However, there was a considerable number of particles between 1 and 10 µm which are responsible for the large effect seen on the PM10 mass distribution. This effect can be observed in Fig. 1c where the normalized mass distribution versus the size distribution is presented.

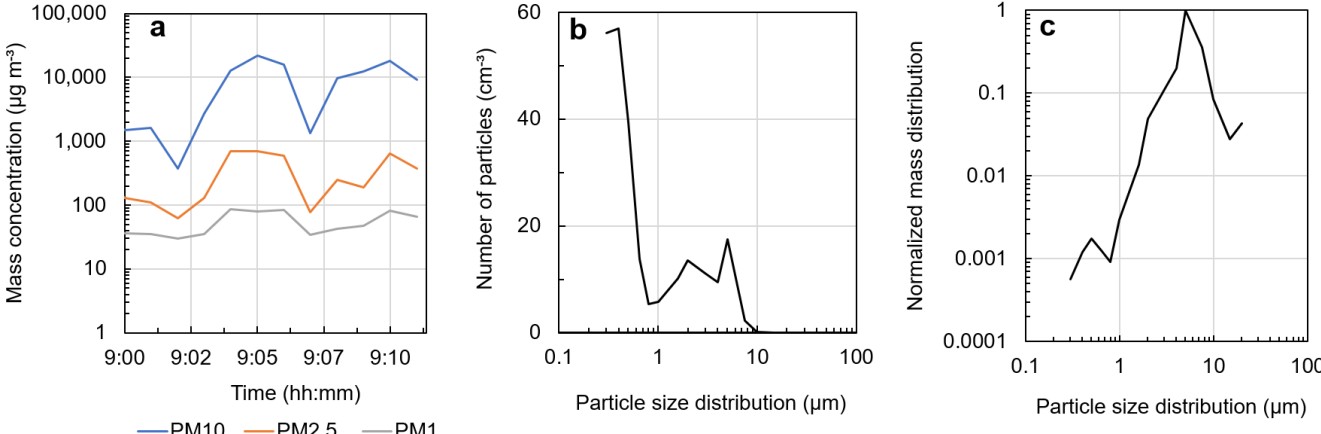

**Figure 1. (a)** Time series of the mass concentrations, **(b)** particle size distribution, and **(c)** normalized mass concentration as a function of the particle diameter during a fog event in Stuttgart (Germany) on 23 January 2020.

Hygroscopicity is an aerosol property that measures its ability to attract and hold water molecules in the condensed phase and determines the variations of aerosol size, and physical and optical properties with RH (Boucher, 2015). The hygroscopic growth factor ($g$) is defined as the ratio between the diameter of the particle at a certain RH and the diameter under dry conditions (Laskina et al., 2015). The hygroscopic growth factor follows a hysteresis (Wise et al., 2005; Li et al., 2014): increasing the RH, one observes a sudden change in the size of the hygroscopic particle due to water uptake. The RH at which this change happens is called the deliquescence point (DRH). Up to this point, a further increase in the RH increases the diameter of the particle, as shown in the study carried out by Wise et al. (2005). If the RH decreases from this point, the particles constantly lose water until the efflorescence point (ERH), where a sudden loss of water and, consequently, a sudden reduction of the size of the particles back to the size under dry conditions occurs.

A lot of research has been done to study the influence of RH on sensor readings (Holstius et al., 2014; Gao et al., 2015; Wang et al., 2015; Jayaratne et al., 2018). However, most of the studies do not differentiate between the growth of hygroscopic particles and fog droplets being detected as particles. Only Jayaratne et al. (2018) investigated both effects separately and raised the question of whether it was possible to correct the particle number and mass concentrations reported by the low-cost sensors in the presence of high humidity and fog.

In Table 1, some of the possible methods to avoid the negative effect of high RH as well as their main advantages and disadvantages are listed. Some research groups have tried to reduce the overestimation of the PM concentrations when the RH is high by using a correction factor based on the κ-Köhler theory (Di Antonio et al., 2018; Crilley et al., 2018). The outcomes show that by applying this correction factor, good results for in situ measurements can be obtained. However, the re-location of the sensors in other places where they are exposed to new environments with different particle compositions limits the transferability of the method. Regression models containing the RH as an independent variable are widely used (Badura et al., 2019; Venkatraman Jagatha et al., 2021; Hong et al., 2021). Nevertheless, researchers indicate the concentration range and specific ambient conditions at which the calibration was performed; for any other conditions, a good performance cannot be

guaranteed. Machine/deep learning techniques are nowadays the most advanced methods in sensor calibration. These computer-based models can potentially be used to correct meteorological effects, cross sensitivities, and sensor drifts (Wang et al., 2020; Kumar et al., 2020). However, they also have limitations such as the high dependency on the quality (accuracy of all input variables, outlier detection) and length of the training data, and the extensive computational resources required.

**Table 1.** Review of possible methods to avoid the negative effect of high RH on sensor readings.

| Methods | Advantages | | Disadvantages | | References |
|---|---|---|---|---|---|
| κ-Köhler theory | • | Consistent results if particle composition is known and constant | • | A change in air masses may lead to over- or underestimations | • (Crilley et al., 2018; Di Antonio et al., 2018; Crilley et al., 2020) |
| | • | Fewer resources needed | • | Limited transferability to other locations | |
| Regression models | • | Consistent results within the calibration range | • | Data extrapolation may lead to wrong results | • (Badura et al., 2019; Hong et al., 2021; Barkjohn et al., 2021) |
| | • | Relatively simple | • | Lack of sensitivity | |
| Machine/Deep Learning | • | Multiple options for algorithms possible | • | Performance depends on the quality of the training data | • (Zimmerman et al., 2018; Wang et al., 2020; Si et al., 2020) |
| | • | Practical for large-scale deployments | • | Limitations to predict uncommon events | |
| | | | • | Extensive computational resources | |
| Diffusion dryers | • | Minimal cost of construction and use | • | Regeneration needed | • (Masic et al., 2020) |
| | • | No energy consumption | • | Not suitable for long-term measurements | |
| Nafion™ membrane | • | No or little maintenance | • | A vacuum system or a drying agent is needed | • (Cai et al., 2014; Karali et al., 2021) |
| | • | Acceptable size and shape | • | Expensive | |
| Thermal drying | • | Drying efficiency variable | • | Excess heating could evaporate volatile and semi-volatile species | • (Samad et al., 2021; Laquai and Kroseberg, 2021; Di Antonio, 2021) |
| | • | Low construction costs | | | |

The pre-conditioning of the inlet air is not a new method. Federal Equivalent Method (FEM) instruments are usually equipped with drying systems like Nafion™ membranes, diffusion dryers, or thermal dryers. The use of Nafion™ membranes is not very popular in the field of PM sensors, most likely because it makes the sensor system incompatible with the term "low-cost" due to its high price. In the case of diffusion dryers, the regeneration process of the silica gel is the main disadvantage as it makes difficult their use in continuous measurements. In this context, a heated inlet appears to be the most reasonable air pre-treatment method. Samad et al. (2021) investigated a low-cost dryer for a medium-cost sensor, the OPC-N3 from the company Alphasense (UK). Laquai and Kroseberg (2021) studied the effect of a low-cost dryer in a cheap PM sensor, the SDS011 from the company Nova Fitness (China), which is a nephelometer. Therefore, we propose to apply a low-cost, thermal dryer as an air pre-conditioning method for the sensor OPC-R1, an optical particle counter from the company Alphasense (UK). Its cost of approximately 100 € makes this sensor an ideal candidate for applications where a certain level of accuracy is expected and

a lot of sensors are needed with a limited budget, for instance in sensor networks for supplemental monitoring or in epidemiological studies.

     The aim of this study was to evaluate a prototype of a low-cost dryer built for a low-cost OPC under two different scenarios, namely fog events and hygroscopic growth, to reduce the influence of RH on the PM readings, i.e. to obtain "reference-equivalent" PM readings. For that purpose, experiments simulating both scenarios were performed under laboratory conditions

and we quantified the effect of the dryer compared to a FEM monitor and a low-cost OPC without a dryer. Additionally, two field campaigns were carried out with the aim of testing the prototype under real atmospheric conditions. In phase I, measurements with the gravimetric reference method, a continuous FEM monitor, and an OPC with dryer were performed in an urban background with daily averages of RH between 70 – 90 %. In phase II, measurements during a fog event (100 % RH) were carried out and the results of the OPC with dryer were compared to a continuous FEM monitor and a sensor without

dryer. Moreover, it was also evaluated whether the use of the low-cost dryer would allow a sensor calibration using exclusively a univariate linear regression (ULR) against gravimetric measurements, without the need for additional variables like the RH.

## 2 Methodology

### 2.1 Instrumentation

     The low-cost dryer was evaluated using an optical particle counter from the company Alphasense, model OPC-R1. For a

detailed analysis of the OPC-R1 performance, we refer the reader to the evaluations carried out by Bulot et al. (2020) and Demanega et al. (2021). The OPC-R1 can measure particles ranging from 0.35 up to 12.4 µm in 16 channels (Alphasense Ltd., 2019). The mass concentrations were directly obtained from the PM outputs of the sensor.

     For the laboratory experiments, the Fidas® 200 was chosen to act as a reference due to its Intelligent Aerosol Drying System (IADS). This instrument has a measuring range covering from 0.18 to 18 µm in 64 channels (Palas GmbH, n.d.). The mass

concentrations were directly obtained from the instrument using the "PM-Ambient" algorithms provided by the manufacturer. The IADS is an air pre-conditioning system consisting of a thermal dryer that is controlled using the temperature and the RH data from an external weather station. It is 1.2 m long and has an inner and outer diameter of 12.7 and 48 mm, respectively. One advantage of the IADS is that it allows the user to work in "expert mode", where the user can decide the heating temperature.

For phase I of the field experiments (daily averages of RH between 70 – 90 %), automatic sequential particulate samplers MicroPNS Type LVS16 with sampling heads for PM10 and PM2.5 from the company MCZ Umwelttechnik (Germany) were used. The air was sampled using filters of 47 mm at a constant volumetric flow rate of 2.3 $m^3\ h^{-1}$. The sample filters were conditioned and weighed according to EN 12341. The PM concentrations were calculated by dividing the net mass gained on the filters by the total air sampled volume. Additionally, a continuous light scattering PM monitor from the company GRIMM

GmbH model EDM 180 was also deployed in the field, together with the sequential samplers and the sensors. In contrast to the Fidas® 200, this monitor integrates a Nafion™ dryer to remove the excess humidity without the danger of losing semi-

volatile organic compounds. In Table 2, the technical specifications of the OPC-R1, the Fidas® 200, and the EDM 180 are presented. For phase II of the field experiments (fog event), the Fidas® 200 was used as a reference.

**Table 2.** Technical specifications of the devices OPC-R1 (Alphasense Ltd., 2019)), Fidas® 200 (S. Hogekamp, pers. comm.; Palas GmbH, n.d.), and EDM 180 (GRIMM Aerosol Technik GmbH, n. d.)

| Methods | OPC-R1 | Fidas® 200 | EDM 180 |
|---|---|---|---|
| Particle size range (µm) | 0.35 – 12.4 | 0.18 – 18 | 0.25 – 32 |
| Number of channels | 16 | 64 | 31 |
| Total flow rate at 25 °C and 1013 hPa (ml min$^{-1}$) | 240* | 4,800 | 1,200 |
| Laser wavelength (nm) | 639 | 390 – 700 | 660 |
| Scattering angle (°) | Multi-angle | 90 | 90 |
| Refractive index | 1.5 + i0 | Confidential | Confidential |
| Density (g cm$^{-3}$) | 1.65 | Dependent on particle size (for PM Ambient) | Confidential |
| Weight (kg) | 0.027 | 13.8 (incl. IADS) | 20 (incl. sampling pipe) |
| Operational temperature range (°C) | -10 – 45 | 0 – 40 | 4 – 40 |
| Operational humidity range (%) | 0 – 95 (non-condensing) | 0 – 100 | 0 – 95 (non-condensing) |
| Internal data storage | no | yes | yes |
| Max. power consumption including dryer (W) | 10 | 200 | 150 |

*Typical flow rate without low-cost dryer

## 2.2 The low-cost dryer

The low-cost dryer for the PM sensor model OPC-R1 consists of a brass tube of 50 cm in length, with an inner and outer diameter of 9 and 10 mm, respectively. The inner diameter was chosen so that the sampling flow rate did not deviate more than 2% from that measured without the dryer. The pressure drop within the tube was estimated to be less than 1.15 Pa considering laminar flow and the properties of air. To build the dryer, ceramic tape is first pasted onto the brass tube to facilitate heat distribution. Next, a wire with a conductor resistance of 0.975 $\Omega$ m$^{-1}$ is wound around leaving 5 cm on each side for ease of handling. To achieve a target power of 10 W with 12 V, 10 windings per cm are needed. In order to attach the dryer to the sensor inlet, the tube was soldered to a copper plate and fixed at the sensor with screws. As it is shown in Fig. 2, the dryer is placed in a vertical position to minimize particle losses. Another important part of the dryer is the insulation. Here, three layers of Thermolam 272 material (100% polyester) are used and the insulated dryer is placed inside a PVC tube as shown in Fig. 2a. The total cost of the material for the construction of the low-cost dryer was approximately 50 €.

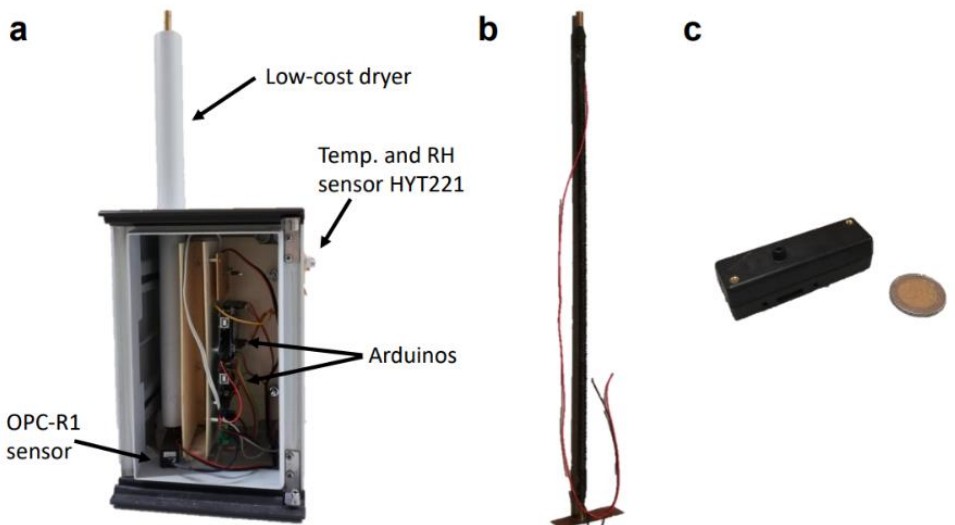

**Figure 2. (a)** Sensor box with low-cost dryer, **(b)** low-cost dryer without isolation, and **(c)** OPC-R1 sensor.

The dryer is controlled by an Arduino Uno microcontroller using the RH data of an ambient temperature and RH sensor, model HYT221 from iST (Switzerland), and the temperature sensor inside the OPC-R1 ($T_{OPC}$). The temperature sensor is located in the OPC circuit board. During the design phase, adding a temperature and an RH sensor at the end of the dryer was considered, but it was discarded as it would have affected the particle flow. The main advantage of using $T_{OPC}$ is the fact that it forms part of the OPC-R1, i.e. the $T_{OPC}$ data is part of the output of the sensor. However, the disadvantage is that using $T_{OPC}$ does not prevent sample overheating. Previous experiments showed that the OPC-R1 switches automatically off if $T_{OPC}$ reaches 44 °C so we selected an upper limit for $T_{OPC}$ of 35 °C.

The Arduino Uno controls the heating using a loop: if the RH is greater than or equal to 65 %, an electrical current will be passed through the wire resistance so that the dryer will be heated. In the second step, the temperature inside the OPC-R1 is used to control the heater. If $T_{OPC}$ is greater than or equal to 35 °C the dryer switches off and starts cooling down to avoid overheating the sensor. Once $T_{OPC}$ is less than or equal to 34 °C and the RH is still greater than or equal to 65 % the dryer will be switched on again.

## 2.3 Laboratory experiments

The experiments were performed in a particle chamber. A schematic set-up of the particle chamber is presented in Fig. 3. The chamber was made from greenhouse glass with aluminium frames and had the following dimensions: 2.57 m long, 1.93 m wide, and 1.95 m high in the middle/highest point. Two OPC-R1 sensors, with and without a dryer, as well as a professional light scattering aerosol spectrometer, model Fidas® 200 from the company Palas GmbH (Germany), were placed in the middle of the chamber. Additionally, two fans were used inside the particle chamber to make sure that the particles were homogeneously distributed.

The experiments to evaluate the dryers under hygroscopic growth conditions were carried out with the help of an atomizer, model 3073 from TSI (US), which generates hygroscopic aerosols from solutions. For that purpose, 80 g l$^{-1}$ solutions of the following pure salts or their mixtures were atomized at 400 hPa: sodium chloride, potassium chloride, ammonium sulphate, and ammonium nitrate. For the experiments with fog, an ultrasonic air humidifier, model U350, from the company Boneco (Switzerland) was used. According to the manufacturer, it produces water droplets with a diameter of up to 4 µm. This model

of humidifier integrates a filter unit (250 AQUA PRO) that allows the generation of water droplets with a lower concentration of impurities than compared without a filter. The impurities in tap water consist mainly in calcium, magnesium, and sodium, which are responsible for the characteristic "white dust" generated by ultrasonic humidifiers (Sain et al., 2018). Moreover, fluoride, nitrate, phosphate, sulphate, aluminium, copper, and iron, among other species, can also be found in different quantities depending on the water quality (Yao et al., 2020; Lau et al., 2021). These impurities act as condensation nuclei

retaining part of the water in the liquid phase, just as fine, suspended particles do during the fog formation in ambient air.
In the first experiments, it was observed that reaching RH higher than 65 % happened slowly when using only the atomizer or the ultrasonic air humidifier. Moreover, the number of particles generated was very high, thus increasing the chances of coincidence errors in both the sensors and the reference instrument. A coincidence error means that there are too many particles in the sensing volume at the same time so the device is not able to resolve every single particle. There is an overlapping of the

single particle signals which causes an underestimation of the particle number concentration and an overestimation of the particle size and, consequently, of the particle mass concentration. Therefore, coincidence errors need to be avoided. To solve this problem, wet towels were used to increase the RH quickly without increasing the number of particles.

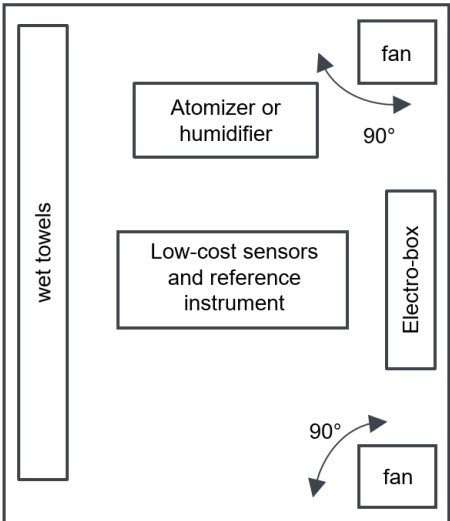

**Figure 3.** Schematic set-up of the particle chamber.

To quantify the effect of the dryer in laboratory conditions, two different drying efficiencies ($\eta_r$, $\eta_s$) were calculated in order to compare the PM2.5 concentrations of the sensor with the low-cost dryer to the PM2.5 concentrations of the reference instrument which also has a dryer (Eq. (1)) and also to the PM2.5 concentrations of the sensor without dryer (Eq. (2)),

$$\eta_r \ (\%) = \frac{\sum_{i=1}^{n} \left( \frac{PM2.5_{d,i}}{PM2.5_{r,i}} \right)}{n} \cdot 100, \tag{1}$$

$$\eta_s \ (\%) = \frac{\sum_{i=1}^{n} \left( 1 - \frac{PM2.5_{d,i}}{PM2.5_{s,i}} \right)}{n} \cdot 100, \tag{2}$$

where $PM2.5_{d,i}$ is the PM2.5 concentration of the sensor with the low-cost dryer at a specific time $i$, $PM2.5_{r,i}$ correspond to the PM2.5 concentration of the reference instrument at a specific time $i$, and $PM2.5_{s,i}$ is the PM2.5 concentration of the sensor without the low-cost dryer at a specific time $i$ for $n$ number of samples. Each drying efficiency provides different information. The $\eta_r$ gives an idea of how close the average PM2.5 readings are between the reference instrument and the sensor with low-cost dryer. In other words, the higher the $\eta_r$ the closer the PM2.5 to "reference-equivalent" PM2.5 readings. The $\eta_s$, in contrast, helps to estimate the actual drying capacity of the low-cost dryer. In the experiments with the air humidifier, it is possible to estimate with $\eta_s$ the ability of the low-cost dryer of removing water from the sample flow. In the case of the experiments with hygroscopic salts, $\eta_s$ estimates the ability of the low-cost dryer to avoid hygroscopic growth.

The time used for determining the dryer efficiency corresponds to the period of time between switching the dryer on and switching the dryer off. To better compare the drying efficiencies, the one-minute averages of the PM2.5 concentrations of both OPC-R1 sensors were corrected by applying ULR against the reference instrument under low RH (the low-cost dryer and IADS dryer were off). The coefficient of determination ($R^2$) was higher than 0.90 in all cases.

## 2.2 Field measurements

The field measurements were performed in two different scenarios (phase I and phase II) to test the dryer under high RH conditions and foggy conditions.

For phase I (daily averages of RH between 70 – 90 %), a measurement station equipped with two sequential samplers, one EDM 180 and one OPC-R1 with a low-cost dryer was deployed in the vicinity of a busy road in Stuttgart (48° 45' 55.8936" N, 9° 10' 12.9396" E) in the period from 21 October 2019 to 5 December 2019, when higher concentrations of ammonium nitrate, which is highly hygroscopic, are expected. Nineteen filters were collected for both PM10 and PM2.5 concentrations. The filters were exposed for 3 days (in the period from 21 October 2019 to 1 November 2019) or 2 days (in the period from 6 November 2019 to 5 December 2019). Data from an OPC-R1 sensor without a dryer was not available for phase I. The data of the sensors and the EDM 180 were averaged to match the gravimetric analysis.

The PM raw data of the sensor with dryer and the FEM instrument were corrected using ULR as shown in Eq. (3),

$$PM_{x,corrected} = \beta_0 + \beta_1 \times PM_{x,raw} \tag{3}$$

where $x$ refers to PM2.5 or PM10, and $\beta_0$ and $\beta_1$ are the calibration constant and the calibration factor of the linear fitting between the sensor or the FEM monitor against the gravimetric measurements, respectively.

For phase II (RH approx. 100 %, fog episode), an OPC-R1 with a low-cost dryer, an OPC-R1 without a dryer, and a Fidas® 200 were collocated at the university campus (48° 45' 1.7316" N, 9° 6' 31.8204" E), a suburban area in Stuttgart-Vaihingen.

The measurements were carried out on the night of 25 January 2022, when a fog event occurred. The PM10 and the PM2.5 concentrations of all the instruments were averaged every minute.

The performance evaluation methods used for field measurements include the standard deviation (SD), slope and offset of the ULR, coefficient of determination ($R^2$), Pearson coefficient (r), Mean Absolute Error (MAE), Root Mean Square Error (RMSE), and Mean Bias Error (MBE). The formulas to calculate the above-mentioned metrics are summarized in Table S1 in the supplemental material.

## 3 Results and discussion

### 3.1 Laboratory experiments

### 3.1.1 Experiments with an ultrasonic humidifier

Experiments with an ultrasonic air humidifier were carried out in the particle chamber to test the efficiency of low-cost dryers to remove water droplets. Figure 4 shows the calibrated PM2.5 concentration of two OPC-R1, with the dryer (red line) and without the dryer (blue line) for two different experiments: in Fig. 4a, the IADS of the reference instrument was kept in automatic mode, i.e., the default settings under which the instrument works during field measurements, whereas in Fig. 4b, it was set at 70 °C using the expert mode. The PM2.5 readings of the reference instrument (black line) are shown for comparison. The comparison with the reference instrument running in automatic mode shows how close the OPC with dryer is at getting "reference-equivalent" PM readings whereas comparing it with an OPC without dryer helps to quantify the amount of water that the dryer can actually remove. In the secondary axis of Fig. 4a and Fig. 4b, the RH (blue dots), as well as the time when the low-cost dryer was on (green line), can be observed.

As shown in Fig. 4a during the experiment with the IADS in automatic mode, once the air humidifier was on, tiny water droplets containing impurities were generated. The water droplets evaporated quickly and, as a consequence, the RH started to increase, leaving the solid impurities with associated water as suspended particles in the air. After the reference instrument reached a PM2.5 mass concentration of 300 µg m⁻³, the air humidifier was switched off. However, the increase in RH was still not enough to start the dryer and wet towels were used to reach an RH higher than 65 %. Immediately after that, a remarkable increase in the PM2.5 concentration was observed, possibly due to the water uptake of the impurities. Once the RH reached 65 %, the low-cost dryer of the OPC-R1 started heating and a pronounced decrease in the PM2.5 concentration is observed, probably due to not only the evaporation of the water but also the evaporation of semi-volatile species. The mean drying efficiencies $\eta_s$ and $\eta_r$ were 64 ± 13 % and 52 ± 10 %, respectively. The reference instrument did not completely evaporate the water and behaved similarly to the OPC-R1 without a dryer. This result was expected, as the Fidas® 200 under default settings does not aim to completely dry the sampled air, but seeks to meet the requirements for FEM instruments as set in the EU directive 2008/50/EC. These requirements are met when the PM readings of the FEM instrument correspond to the values of the measured PM filters of the standard gravimetric analysis after being pre-conditioned at 19 to 21 °C and 45 to 50 % RH for

at least 48 h (EN 12341). For that reason, and in order to have PM results as close as possible to the gravimetric measurements, the heating power used by the IADS was less than 25 % of the total power (90 W) during the experiment. The IADS regulates the heating considering the RH in the air, which in this experiment did not reach more than 75 %, and therefore, the IADS considered sufficient a heating power of less than 25 %.

In order to determine an "apparent temperature" of the low-cost dryer, experiments in expert mode varying the temperature of the IADS were performed and the closest result is presented in Fig. 4b, in which the IADS was set using the expert mode at 70 °C. This "apparent temperature" is not the real temperature of the dryer as it was designed to keep a constant heat flux (through electric heating) and therefore the dryer has a temperature profile that varies through the length. More information about the air temperature inside the dryer has been summarized in section 3.1.3.

In Fig. 4b, the evaporation of water and semi-volatile species was clearly observed for both the reference instrument and the OPC-R1 with a dryer, reaching the latest a mean drying efficiency ($\eta_s$) of 57 ± 13 % compared with the OPC-R1 without a dryer. The mean drying efficiency with respect to the reference instrument ($\eta_r$) is in this case 84 ± 15 %. However, this number does not really indicate how close the sensor with low-cost dyer is to achieving "reference equivalent" PM readings, as the Fidas 200 was not working under default settings (automatic mode). Further information about the temperature of IADS during

the experiments with the ultrasonic humidifier in the laboratory can be seen in Fig. S1 and S2 of the supplemental material.

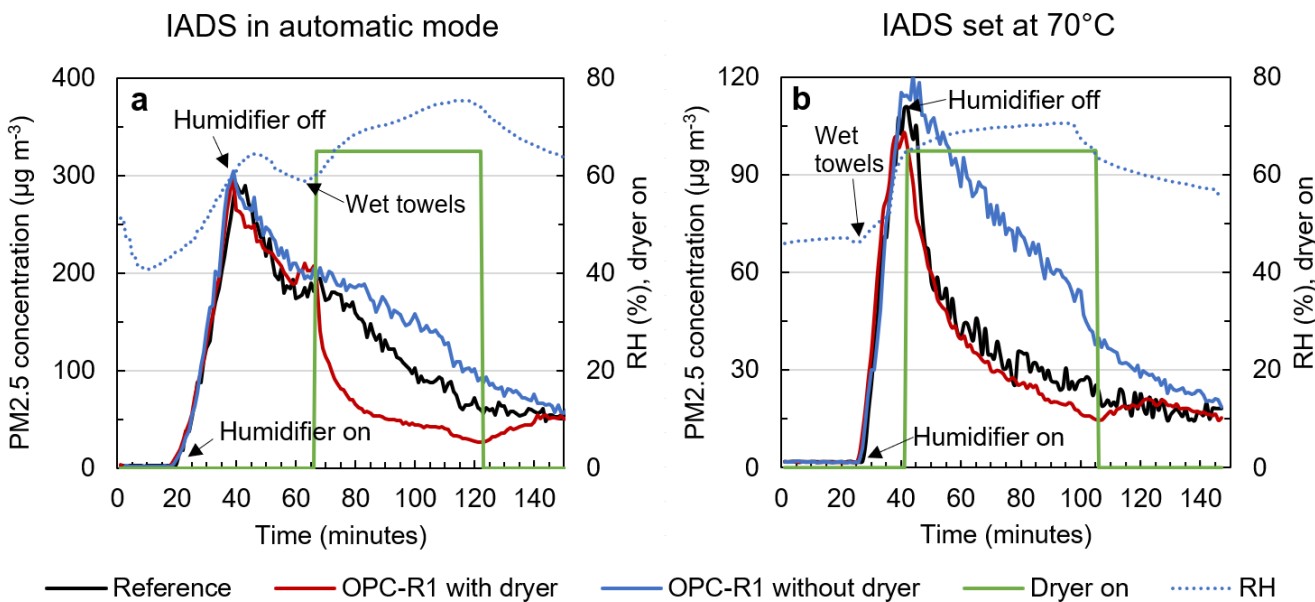

**Figure 4.** Experiments with an air humidifier **(a)** keeping the IADS in automatic mode and **(b)** IADS set at 70 °C.

Figure 5 illustrates the size distribution of the PM generated with the ultrasonic air humidifier measured with the reference instrument. As can be seen, the mean diameter was below the detection limit of the reference instrument (0.18 µm) and the

OPC sensors (0.35 µm). As shown in Fig. 1c, fog events in the field have a different size distribution with particles ranging also from 1 to 10 µm. Another limitation that was found during these experiments is the fact that it was not possible with the

proposed set-up to reach RH close to 100 % without having coincidence errors. Therefore, for future research with fog droplets, other types of fog generation like the ones suggested by Angelov et al. (2017) but also field measurements in real fog conditions are recommended.

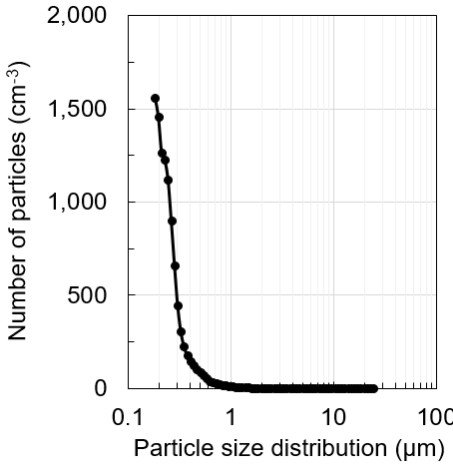

**Figure 5.** Particle size distribution measured by the reference instrument during the experiments with the ultrasonic humidifier.

These experiments demonstrate the positive effect of the low-cost dryer to remove water droplets and hence decrease the overestimation of the PM2.5 concentration during fog events. The energy needed to remove the water is significant and even the reference instrument is not able to remove all the water when working in automatic mode. This outcome is similar to that reported by Jayaratne et al. (2018) who wrote "The corresponding increase in the TEOM reading…suggests that, in the presence of fog, the dryer at its inlet has a limited efficiency in terms of removing the liquid phase of the particles". The WMO/GAW guidelines recommend modest heating so that sampled air temperature does not exceed 40 °C to minimize the loss of semi-volatile species (WMO/GAW, 2016). However, the findings from these experiments suggest that temperatures higher than 40 °C are needed in order to observe a clear reduction of the mass concentration during fog events. Consequently, an optimum has to be found between the efficient removal of fog and the minimization of the loss of semi-volatile species. This has special implications in regions where fog formation is abundant in terms of probability, frequency, and duration. One possible solution is introducing adaptive heating to the dryer control to keep the RH of the air at the sensor inlet constant at 50 %. In such a case the temperature needed to maintain the RH of the air at 50 % could be adjusted so that higher temperatures than 40 °C would only be reached during fog events, where the RH is close to 100% in order to be able to counter-react the effect of the fog in the PM readings. As can be seen in Fig. S9 in the supplemental material, the IADS of the Fidas® 200 also reached temperatures higher than 40 °C (51 to 53 °C) during the real fog event.





### 3.1.2 Experiments with hygroscopic aerosols

Experiments were carried out with different aerosols (($NH_4$)$_2SO_4$, $NH_4NO_3$, KCl, and NaCl) and different IADS settings (automatic mode, IADS off (20 °C), 35 °C, 50 °C, and 65 °C). Figure 6 shows the results of an experiment carried out to test the dryer against hygroscopic growth with ($NH_4$)$_2SO_4$ particles. For this experiment, the Fidas® 200 ran in automatic mode. Experiments with $NH_4NO_3$ and the mixture of the salts can be seen in Fig. S3 and S4 of the supplemental material, respectively. The DRH and ERH of ($NH_4$)$_2SO_4$ as well as the other tested salts are indicated in Table S2 in the supplemental material. Once constant concentrations were reached in the particle chamber, wet towels were introduced to increase the RH quickly. The effect of the sudden increase in the RH can be clearly seen at minute 45 in Fig. 6a by the simultaneous increase in the PM2.5 concentration in all the devices. As soon as 65 % RH is reached, the dryer switched on automatically and after one minute the PM2.5 concentration measured by the OPC-R1 with the dryer drastically decreased. A decrease was also observed with the reference instrument but at a slower pace. This was due to the reaction time of the RH sensor that controls the IADS of the reference instrument (in brown dots in Fig. 6a) which reacts slower compared to the RH sensor (blue dots in Fig. 6a) that controls the low-cost dryer. Consequently, the IADS increased the heating power much more slowly. This decrease in the PM2.5 concentration of the reference was observed between minute 46 and minute 60 after the wet towels were introduced into the particle chamber. However, this was also observed in the OPC-R1 without the dryer as well as in the OPC-R1 with the dryer, which means that the decrease could have other causes, for instance, the sedimentation of the heavier particles or particle deposition onto the wall. From minute 60 until the end of the experiment the PM2.5 concentration of the reference instrument did not vary significantly.

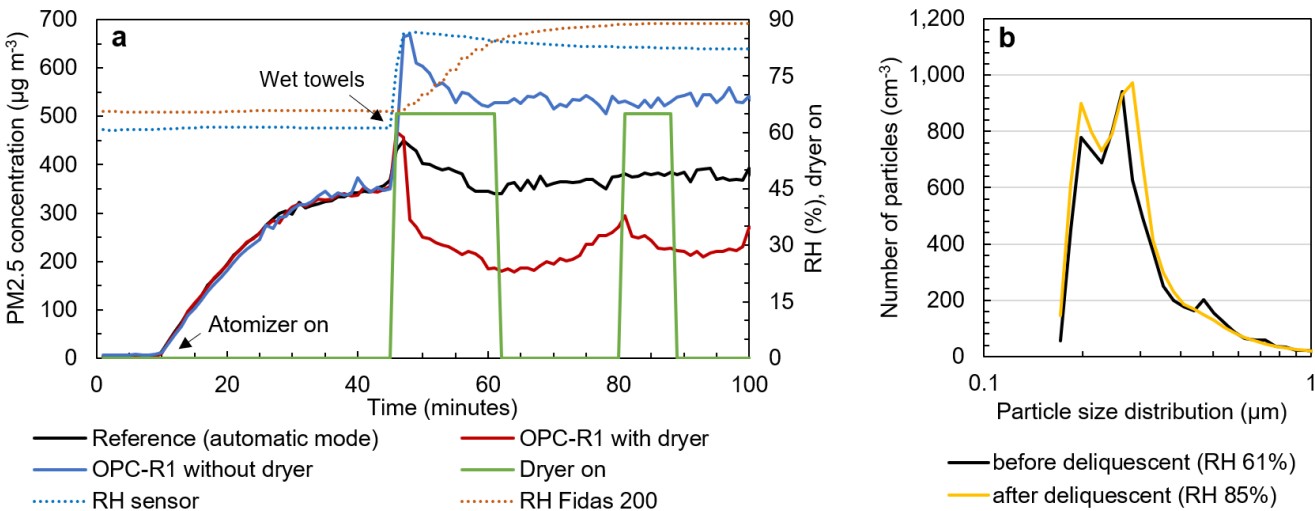

**Figure 6. (a)** Time series of the PM2.5 concentration during an experiment with ($NH_4$)$_2SO_4$ particles, **(b)** particle size distribution measured by the reference instrument before and after deliquescence.

The drying efficiency of the low-cost dryer, when compared with the reference instrument ($\eta_r$), was 63 ± 5 %, whereas it was 57 ± 4 % when compared with the OPC-R1 without dryer ($\eta_s$), including both periods when the dryer was on. An interesting

observation is that in the periods when the low-cost dryer was switched on (marked with a green line in Fig. 6a), the PM2.5 concentration measured by the OPC-R1 with the dryer decreased and increased again when the dryer was switched off. This pattern is not observed in the reference instrument, whose PM2.5 concentration readings remained constant at around 380 µg m⁻³. Further information about the IADS temperature during this experiment is shown in Fig. S5 in the supplemental material.

In Fig. 6b it can be observed that the reference instrument almost completely avoided the shifting of the curve to the right after the deliquescent point when comparing the particle size distribution of the $(NH_4)_2SO_4$ particles before and after the deliquescent point. It should be also highlighted that approx. 80 % of the particles seen by the Fidas® 200 have a mean diameter from 0.17 to 0.35 µm, which means that the OPCs are not detecting a substantial amount of material.

An experiment with $NH_4NO_3$ particles is shown in Fig. S3, in which different IADS temperatures were manually set (20 °C, 330 35 °C, 50 °C, and 65 °C). This experiment clearly shows the impact of temperature in the loss of semi-volatiles due to evaporation and, therefore, the detrimental effect that the use of high temperatures in a heated inlet can have on the mass concentration when species with high volatility like $NH_4NO_3$ are present in the sample. In this sense, the presented design of a low-cost dryer is behaving as a thermodenuder, i.e., a device that is used to study the volatility fraction of aerosol particles (Huffman et al., 2008). Studies using thermodenuders have shown that temperatures of 83 to 88 °C can cause 50 % of the 335 organic aerosol mass to evaporate (Paciga et al., 2016). For the specific case of nitrate, much lower temperatures are needed to reduce the mass by 50 %, as it is shown in the results of Huffman et al. (2009) where 50 % of the nitrate during a field campaign was evaporated at 54 °C.

### 3.1.3 Study on the drying temperature

To get more information about the temperature profile inside the dryer, experiments were performed in the laboratory where 340 the temperature of the air flowing inside the dryer was measured. The experiments showed that the maximum wall temperature is reached at 40 cm (Fig. S6). In the last centimeters, the air is cooled down before the sensor inlet due to the lack of heated wire (the last 5 cm were left wire-free for ease of handling). It was observed that at 40 cm the air is heated up to approx. $65.9 \pm 0.5$ °C. This is in agreement with the experiments which show that the sensor with low-cost dryer behaves similarly to the reference instrument if the IADS is heated at 70 °C. As the thermocouple influences the airflow, the measured temperature 345 may have some bias, but it is clear that it is higher than 40 °C, which is the maximum temperature recommended by the WMO/GAW guidelines for ambient air monitoring. Moreover, it was observed that the $T_{OPC}$ is usually $10 - 13$ °C higher than the ambient temperature, which means that the dryer may not start heating when the ambient temperature is higher than $22 - 25$ °C, as the $T_{OPC}$ could be already higher than the temperature limit set for $T_{OPC}$ (35 °C). This problem could be solved by changing the upper limit temperature loop in the Arduino code. However, this change also increases the maximum air 350 temperature in the dryer, which is already too high for producing "reference-equivalent" PM readings. Therefore, we recommend that new versions of the low-cost dryer should focus on the control of the RH in the sample flow, as the $T_{OPC}$ value is highly dependent on the ambient air temperature.

## 3.2 Field measurements

Field measurements were performed to evaluate the effect of the low-cost dryer under two different scenarios: hygroscopic growth (phase I) and fog conditions (phase II).

### 3.2.1 Field measurements in a period with high relative humidity

The results of phase I for PM10 and PM2.5 are presented in Fig. 7. The aim of phase I was to compare the OPC-R1 with low-cost dryer against gravimetric measurements, and a FEM monitor EDM 180 in an urban background in a period with high RH (daily averages RH between 70 – 90 %). To evaluate if the sensor data with a low-cost dryer could be corrected with a univariate linear regression (ULR), the data was divided into two sets: the first data set (from 21 October 2019 to 17 November 2019) was used to calibrate the sensor data with a ULR compared to the gravimetric measurements, whereas the second data set (18 November 2019 to 5 December 2019) was used to evaluate the performance of the applied ULR. The same calibration procedure was done with the data of the FEM instrument (EDM 180).

During phase I, the daily average of the RH was between 76 to 86 %. Due to the temperature control loop, the dryer was not on continuously, but only part of the time, as it is indicated in the secondary axis in Fig. 7a and 7b. The analysis of the raw data in Fig. 7 shows a significant difference between the behavior of the PM10 and the PM2.5 concentrations compared to the gravimetric analysis. Whereas the sensor with low-cost dryer tends to overestimate the PM10 raw data, the PM2.5 is frequently underestimated. This underestimation occurs probably due to two reasons: (1) most of the semi-volatile organic compounds belong to the PM2.5 fraction and the dryer could be evaporating them and (2) the lower limit of the particle size in an OPC-R1 is 0.35 μm and a significant number of particles in the urban background are smaller than that. It is likely that reason (1) prevails over (2), as a significant amount of ammonium sulphate and, especially, ammonium nitrate, is expected in the PM2.5 fraction. On the contrary, the EDM 180 tends to underestimate PM10 and overestimate PM2.5.

In Table 3, the summary of statistics for PM10 is presented. Based on results for the PM10 raw data, the EDM 180 shows a higher correlation to gravimetric measurements compared to the sensor with low-cost dryer, having the EDM 180 and the OPC-R1 an $R^2$ of 0.93 and 0.61, respectively. In general, the EDM 180 shows slightly lower errors (MAE, RMSE, and MBE) than the OPC-R1 with low-cost dryer. The slope (1.1) and the offset (1.7) of the OPC-R1 are closer to one and zero, respectively, which favors the use of a ULR to correct the sensor data. In fact, after calibration of the PM10 concentration, the OPC-R1 shows a good agreement with the gravimetric analysis, presenting an even lower MAE (2.4 μg m$^{-3}$) than the calibrated EDM 180 (2.8 μg m$^{-3}$), and a higher $R^2$ (0.9) and r (0.95) than the raw data.

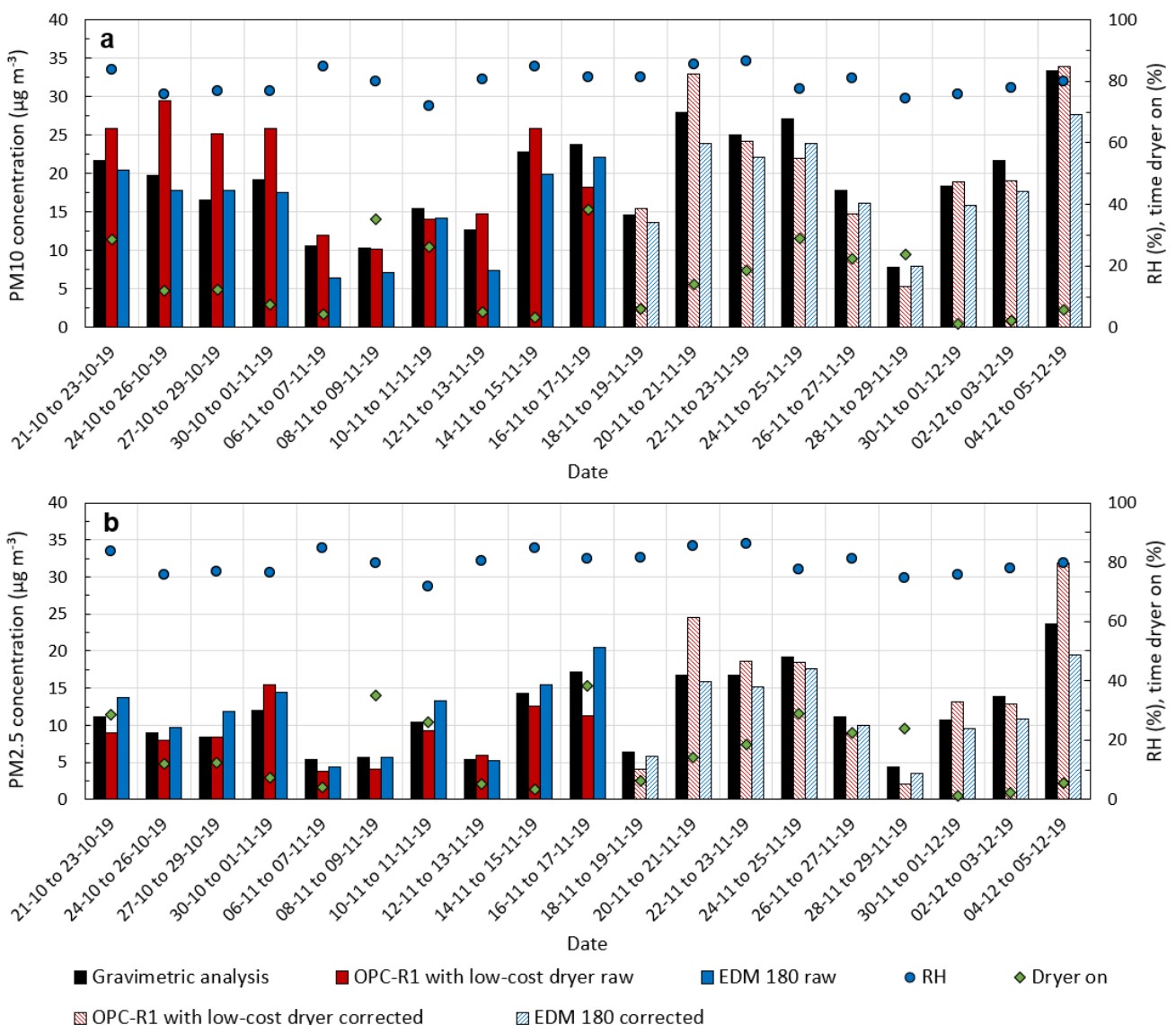

**Figure 7.** Comparison of gravimetric analysis, an OPC-R1 with low-cost dryer, and a FEM monitor EDM 180 for **(a)** PM10 concentrations, and **(b)** PM2.5 concentrations. Solid bars are used for calibration and bars filled with patterns represent corrected data.

As can be seen in Table 4, the raw PM2.5 data of the EDM 180 and the OPC-R1 with low-cost dryer show a similar agreement

with respect to the gravimetric analysis with both having MBE lower than $\pm$ 1.5 µg m$^{-3}$ and MAE lower than 2 µg m$^{-3}$. After

correction with ULR, the EDM 180 does not show any significant improvement, except for a higher $R^2$ and r of 0.98 and 0.99,

respectively. The OPC-R1 with low-cost dryer also improves the $R^2$ and r (0.90 and 0.95, respectively) but shows an increase

in the MAE, the RMSE, and the MBE (3.2, 4.1, and 1.3 µg m$^{-3}$, respectively).

**Table 3.** Summary of statistics for PM10 concentration for the raw data (21 October 2019 to 17 November 2019) and the data calibrated with ULR (18 November 2019 to 5 December 19).

| | SD | MAE | RMSE | MBE | slope | offset | $R^2$ | r |
|---|---|---|---|---|---|---|---|---|
| **PM10 raw data** | | | | | | | | |
| EDM 180 | 5.7 | 2.5 | 2.8 | -2.2 | 1.2 | -5.1 | 0.93 | 0.96 |
| OPC-R1 with dryer | 6.7 | 4.3 | 5.3 | 2.8 | 1.1 | 1.7 | 0.61 | 0.83 |
| **PM10 calibrated data** | | | | | | | | |
| EDM 180 | 5.8 | 2.8 | 3.2 | -2.8 | 0.8 | 1.8 | 0.99 | 1.00 |
| OPC-R1 with dryer | 8.5 | 2.4 | 2.9 | -0.8 | 1.1 | -2.8 | 0.90 | 0.95 |

**Table 4.** Summary of statistics for PM2.5 concentration for the raw data (21 October 2019 to 17 November 2019) and the data calibrated with ULR (18 November 2019 to 5 December 19).

| | SD | MAE | RMSE | MBE | slope | offset | $R^2$ | r |
|---|---|---|---|---|---|---|---|---|
| **PM2.5 raw data** | | | | | | | | |
| EDM 180 | 4.9 | 1.8 | 2.2 | 1.5 | 1.3 | -1.1 | 0.95 | 0.97 |
| OPC-R1 with dryer | 3.5 | 1.9 | 2.5 | -1.1 | 0.8 | 1.3 | 0.68 | 0.82 |
| **PM2.5 calibrated data** | | | | | | | | |
| EDM 180 | 5.1 | 1.7 | 2.0 | -1.7 | 0.9 | 0.2 | 0.98 | 0.99 |
| OPC-R1 with dryer | 9.0 | 3.2 | 4.1 | 1.3 | 1.5 | -5.1 | 0.90 | 0.95 |

In general, the field measurements during phase I have shown that the use of a low-cost dryer in an urban background under high RH conditions may be beneficial to allow the calibration of the PM10 concentrations with a ULR. Special care should be taken when interpreting the results for the PM2.5 fraction. It is difficult to draw a conclusion as the worsening of the metrics after the ULR could have been due to the evaporation of semi-volatile species or since two of the testing data points (from 24 to 25 November 2019 and from 4 to 5 December 2019) were out of the calibration range and extrapolation can be a big source of error when using ULR.

### 3.2.2 Field measurements during a fog event

In phase II, two OPC-R1 with and without dryer were collocated at the university campus during a fog event on the night of 25 January 2022. The data were averaged every one-minute. The results of the raw data for PM10 and PM2.5 concentrations are shown in Fig. 8. The RH measured by the weather station of the Fidas® 200 remained close to 100 % during the whole duration of the fog event. The time when the dryer was on or off is presented in the secondary axis (green dots). The PM10 and the PM2.5 outputs of the sensors have been compared to the Fidas® 200 and a summary of the statistics can be seen in Table 5.

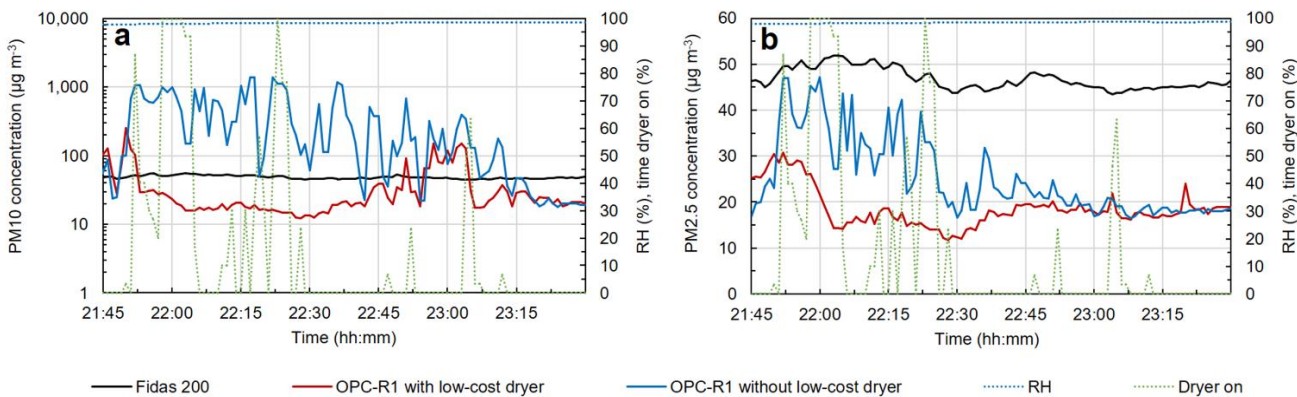

**Figure 8.** Time series of an OPC-R1 with low-cost dryer, an OPC-R1 without low-cost dryer, and a Fidas® 200 during a fog event for **(a)** PM10 concentrations, and **(b)** PM2.5 concentrations.

As can be seen in Fig. 8, the FEM instrument (Fidas® 200) kept the PM10 and the PM2.5 concentrations quite constant with averages of 48.7 and 46.8 µg m$^{-3}$, respectively, and standard deviations lower than 3 µg m$^{-3}$. The PM10 concentrations of the OPC-R1 without dryer show on the contrary a completely different behavior, measuring most of the time PM10 concentrations

in the order of $10^2$ µg m$^{-3}$ with very sharp fluctuations (average 340.4 ± 375.4 µg m$^{-3}$). All the error metrics are extremely high, highlighting an MBE of 291.7 µg m$^{-3}$. It is very clear that the PM10 concentrations of the OPC-R1 without a dryer are affected by fog.

With respect to the PM2.5 concentration, the data measured by the OPC-R1 without dryer remained below the reference instrument (25.2 ± 8.6 µg m$^{-3}$). It can also be seen that both sensors (with and without the dryer) measured the same

concentrations in the periods when the dryer was switched off. In order to explain this behavior, it is important to take into account the particle size distribution during this event. As shown in the Fig. S7 and S8 of the supplemental material, approx. 63 % of the total mass corresponded to particles smaller than 0.35 µm, which is the lower limit of the particle size measured by the OPC-R1. That means that the sensors were not detecting an important number of particles, which probably explains the big difference in the PM2.5 concentration found between the sensors and the Fidas® 200.

The PM10 and PM2.5 concentrations measured by the OPC-R1 with dryer were kept in concentrations lower than the measured by the FEM instrument and the OPC-R1 without dryer when the dryer was on (average 36.7 ± 38.5 µg m$^{-3}$ for PM10 and 18.5 ± 4.2 µg m$^{-3}$ for PM2.5). This occurs, as was also shown during the laboratory experiments, due to the effect of the low-cost dryer, which is not only evaporating the bigger water droplets but also drying completely the hygroscopic aerosols to RH below their ERH, so that the particles are too small to be detected by the sensor.

As can be seen in Table 5, the slope and the offset of the PM10 and the PM2.5 data of the sensor without dryer are far from being close to one and zero, respectively, Therefore, the idea of correcting the data with ULR was discarded. Similarly, the OPC-R1 with dryer shows an $R^2$ and an r of 0.02 and -0.13, respectively, for PM10, and 0.04 and 0.19, respectively, for PM2.5. That implies there is no meaningful relationship between the Fidas 200 and the data of the OPC-R1 with the dryer. One problem

of the presented dryer prototype is the lack of continuity of the drying process, which implies that a constant temperature in a
steady state is never reached, which does not favor the possible use of a ULR for data correction for minute-average values.
As shown in Fig. S9 in the supplemental material, the temperature of the IADS system during the fog event was kept constant
between 51 and 53 °C.

**Table 5.** Summary of statistics for the PM10 and PM2.5 concentration (raw data) during phase II of field measurements (fog event).

|  | Average | SD | MAE | RMSE | MBE | slope | offset | $R^2$ | r |
|---|---|---|---|---|---|---|---|---|---|
| **PM10** | | | | | | | | | |
| Fidas® 200 | 48.7 | 2.6 | | | | | | | |
| OPC-R1 with dryer | 36.7 | 38.5 | 33.1 | 40.7 | -12.0 | -1.8 | 126.3 | 0.02 | -0.13 |
| OPC-R1 without dryer | 340.4 | 375.4 | 301.8 | 474.5 | 291.7 | 61.8 | -2669.4 | 0.19 | 0.44 |
| **PM2.5** | | | | | | | | | |
| Fidas® 200 | 46.8 | 2.3 | | | | | | | |
| OPC-R1 with dryer | 18.5 | 4.2 | 28.2 | 28.6 | -28.2 | 0.4 | 2.0 | 0.04 | 0.19 |
| OPC-R1 without dryer | 25.2 | 8.6 | 21.6 | 22.7 | -21.6 | 2.8 | -105.2 | 0.56 | 0.75 |


## 5 Discussion

The results of the experiments carried out in the laboratory as well as the field campaigns have proven the ability of the dryer
prototype to revert the hygroscopic growth and evaporate the fog before the sensor inlet. Calibration of an OPC-R1 with a
low-cost dryer during periods of high RH (70 – 90 %) by using ULR showed favourable results for the PM10 concentrations.
The calibration of the PM2.5 fraction did not seem to improve the results, but the MBE was kept low (1.3 µg m$^{-3}$). However,
the presented prototype of dryer causes an excess of heating clearly identified when compared to FEM monitors during the
real fog event (phase II) and in the laboratory experiments. It was not possible to correct minute-average sensor data with ULR,
as the dryer is continuously switching on and off. This intermittent on/off process does not seem to be a problem when the
averaging time covers longer periods as shown in the results of phase I.
Due to the higher temperatures reached in the inlet, the risk of evaporating semi-volatile organic compounds exists. Moreover,
the hygroscopic aerosols are fully dried to levels below the ERH. This may deviate the PM concentrations of the sensor with
low-cost dryer from the gravimetric measurements where the filters (and therefore the sampled particulate matter) are kept at
50 % RH. Due to the hysteresis presented in the hygroscopic growth and shrinkage, some aerosols may still contain water at
50 % RH, depending on their ERH.
Due to all the above-mentioned problems, the prototype of low-cost dryer should be further investigated and optimized.
Therefore, adaptive heating aiming for a constant RH of the incoming air of 50 % should be considered for new versions of
the low-cost dryer. Additionally, a temperature limit of 40 °C should be introduced, as recommended by the WMO/GAW

guidelines. Furthermore, keeping a constant temperature instead of constant heat flux could improve the later application of ULR for data correction.

It is worth noting that even though drying the air could raise questions about how the temperature affects the physico-chemical properties of the particulate matter, existing software solutions are not problem-free, as they may fail when changes in the particle chemical composition occur. Moreover, a software solution that helps to minimize the effect of the hygroscopic growth and fog events in the mass concentrations has not been reported in the literature. In general, the effect of fog on the mass concentrations of sensors has scarcely been addressed in the literature.

It is clear that PM sensors have come to the air quality monitoring market to stay, and that (i) all new approaches (hardware, software, or hybrid solutions) aiming at improving the accuracy of PM sensors, and (ii) evaluations on how they behave when the environmental conditions change due to e.g. fog events, long-range transport, or a change in sensor location, are welcome to be addressed in future research.

## 6 Conclusions

Fog events and the ability of hygroscopic aerosols to uptake water can cause an overestimation of the mass concentrations in low-cost sensor readings based on light scattering. Low-cost sensors are already and will be a game changer in the future of air pollution monitoring. Finding a solution for these problems will make the sensor data more accurate, expanding possible application fields to those where a high level of accuracy is required, e.g., in supplemental monitoring or epidemiological studies. The present study provides an overview of the work carried out for the evaluation of a self-constructed, low-cost dryer

for a low-cost optical particle counter under laboratory and field conditions. It was shown that low-cost, thermal dryers can be a cost-effective solution to avoid the negative effect of hygroscopic growth and fog droplets on the mass concentration readings of low-cost optical particle counters. The investigated dryer has been proven to be very effective in reducing the water content of hygroscopic particles or fog. The results also indicate that our prototype dries the particles more than FEM instruments, which suggests that this design of a low-cost dryer is over-dimensioned in terms of heating power to have "reference-

equivalent" PM readings. A comparison with gravimetric analysis has also shown that, under conditions of high RH (70 – 90 %), a ULR could correct the PM10 concentrations of the OPC-R1. For PM2.5 concentrations, the results are not fully satisfactory, but the errors remain low (MBE 1.3 µg m⁻³). The PM10 concentrations of the sensor without a dryer during a fog event in the field showed an important overestimation (factor 7) compared to the FEM instrument. This overestimation was not seen in the sensor with dryer which measured lower PM10 concentrations than the FEM monitor. With respect to PM2.5,

both sensors (with and without a dryer) measured concentrations lower than the FEM instrument. However, this outcome is highly dependent on the ambient particulate matter of the location. As reported by Jayaratne et al. (2018) who measured close to a busy road, the PM2.5 concentrations of sensors can also be largely overestimated during fog periods. Finally, correcting one-minute average PM values of the OPC-R1 with our prototype of low-cost dryer during a fog event was not possible due to the discontinuity of the drying process.

It should be highlighted that low-cost dryers do not eliminate the need for calibration but, because of their simplicity, they are very promising for applications where complex data post-processing is too difficult/expensive, e.g., in citizen science projects. Moreover, the design of the dryer can be easily adapted to other models or types of sensors, including, for instance, electrochemical sensors for gases as tested by Samad et al. (2020).

During the laboratory experiments, some challenges were encountered. Some of these were the impossibility of reaching 100 %

RH in the particle chamber without causing coincidence errors and the difficulties in the generation of water droplets that could simulate the size distribution of real fog. The mean diameter of the generated fog droplets was < 1 µm, whereas fog observed during field measurements and what has been found in the literature have a bigger fraction of droplets between 1 and 10 µm. Another challenge encountered was simultaneously (a) removing fog droplets, (b) minimizing the effect of the hygroscopic growth, and (c) avoiding the evaporation of volatile organic compounds. Furthermore, more research is required to optimize

the air temperature and the energy consumption and to create an adaptive heating based on the real need for heating according to the meteorological conditions to keep the RH of the air in the sensor inlet at approx. 50 % in order to produce "reference-equivalent" PM readings.

**Code availability**

The Arduino codes are available at https://github.com/MiriamChacon/OPC-R1_with-air-dryer (last access: 22 November

2022) and archived on Zenodo (DOI: 10.5281/zenodo.7045960).

**Data availability**

The data of this study are available from the authors upon request.

**Author contributions**

MCM drafted the manuscript. MCM and BL designed the research plan. MCM and BL carried out the experimental work.

MCM performed the data analysis. UV and CS supervised the project. UV secured the funding. BL, UV, and CS provided extensive comments on this manuscript. All authors had read and approved the final manuscript.

**Competing interest**

The authors declare that they have no conflict of interest.

**Acknowledgement**

Our sincere gratitude to Stefan Hogekamp (Palas GmbH) who supported the study with a reference instrument and his knowledge and to Sandra Grob and Clémence Aubert for their contribution to the execution of the experiments. The authors also thank the three anonymous reviewers for their suggestions to improve the quality of the paper.

**Financial support**

This research was funded by the German Environment Agency (project FKZ: 3718-51-240), and by the Ministry for Social
Affairs and Integration Baden-Württemberg (project AZ 53-5425.1/5).

**Review statement**

This paper was edited by Francis Pope and reviewed by three anonymous referees.

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
