# Peer review of "Evaluation of a Low-Cost Dryer for a Low-Cost Optical Particle Counter"

_Atmospheric Measurement Techniques, 2022_

## Referee Comment (RC1)

This work evaluates the performance of a low-cost thermal dryer for the OPC-R1, in comparison to an OPC-R1 without a dryer and the Palas Fidas 200, which has an automated drying system. Two separate scenarios which can lead to erroneously high PM readings at elevated relative humidity (RH) were investigated in a laboratory setting. (1) Fog conditions, where water droplets suspended in the air are incorrectly detected as particles, and (2) hygroscopic growth of particles, leading to elevated size and mass measurement relative to a measurement at lower RH. To evaluate the performance of the R1 with dryer against the other two instruments, both the time series and "drying efficiency" are presented. The authors conclude that the low-cost dryer was able to mitigate positive artifacts due to the two effects outlined above, but raise concerns over the comparability of their "fog" conditions to the real-world scenario. They state the need for further work to characterise the temperature profile and optimise the power requirements of their drying system.

This work is a useful addition to the literature on low-cost particulate matter sensors. The literature review is thorough and highlights that drying systems are both under-studied and potentially simpler to implement than post-calibration of measurements for RH. The two scenarios being investigated are clearly distinguished and explained.

I would like to highlight two main aspects of the paper which I think should be clarified. I think this is good work; my concerns relate to how the results are presented. I think the results could be more clearly placed within the context of standard reference measurements of PM, and certain aspects of the method are unclear.

It is not clear to me what the authors are trying to achieve through use of their low-cost dryer. Multiple comparisons are made which all suggest a different target. Are they trying to (1) completely remove all water, (2) produce "reference-equivalent" PM readings to supplement regulatory monitors, or (3) simply reduce the positive artifact due to the presence of water? I think this needs to be stated from the outset as it affects which evaluation method is most appropriate.

Comparing the R1 + dryer to the R1 without a dryer proves only aim (3), that water has been removed by heating. The reading is closer to dry particles, but we don't know how close.

Comparing the R1 to the Palas may prove (1) or (2). My uncertainty here is due to the lack of information on how the Palas PM measurements were processed. The Palas can output (1) raw particle number size distribution, (2) PM "classic" and (3) PM "ambient". The authors could have obtained PM mass from (1) by applying their own density and penetration factors, (2) is calculated by the Palas using a default assumed density $\rho = 1.3$ gcm$^{-3}$, whilst (3) has an additional empirical correction factor applied to make the readings "reference-equivalent" (Di Antonio 2020).

Firstly, I would be unable to reproduce these findings without knowing which of the three methods the authors have used.

Secondly, if the PM "ambient" readings have been used, I think it is worth pointing out that (under default IADS settings) the Palas is not *trying to fully dry the particles* because it is aiming to be reference-equivalent. Hence this is relevant to aim (2). The EU reference standard specifies conditioning for 24-hr at 20°C, 50% RH (see CEN standard 12341), ie not completely dry. Given this, the Palas can be expected to potentially measure PM which still contains bound water and not remove all fog droplets as it may not sufficiently lower the RH.

Thirdly, when the authors manually set the IADS to 70°C, they were overriding these reference-equivalent settings and were likely removing nearly all the water. This comparison is most appropriate to aim (1).

A short explanation of what each of the authors' comparisons (R1 without dryer, Palas automated settings, Palas at 70°C) actually means in the context of how much of the water is being removed and the relation to regulatory PM monitoring would make this paper more impactful.

A more precise description of how the Palas PM values were obtained is essential to make this work replicable by other authors. There should also be a more detailed description of how the R1 PM values were derived- were they the firmware-output values and what density and refractive index have been used? I would also point out that some discrepancy between R1 and Palas PM readings is to be expected because the R1 has a larger minimum detectable diameter.

My second point of feedback is that the meaning of the $T_{OPC}$ quantity should be clarified. In particular, the authors should specify what this temperature corresponds to- is the sensor situated (1) on the OPC circuit board, or (2) within the sample air flow? The authors should then be clear about what they are preventing from overheating by including an upper temperature limit.

I suspect case (1) applies, from my own work with the OPC-N3, a similar model of OPC. This makes $T_{OPC}$ the temperature of the OPC box/components rather than the sample. Therefore the upper temperature limit of 35 °C would be preventing the OPC from overheating and is not an appropriate control measure to prevent excessive sample heating.

As the authors state in the penultimate line, it is a shame that no information has been provided on the temperature (and RH) that the dryer conditions the sample to. This would be an important parameter in assessing which of the above three aims the dryer is best suited to. From the fact that the IADS had to be set to 70 °C to achieve comparable PM readings, I suspect that the sample is being heated much higher than the 35 °C limit on $T_{OPC}$. This information would also be useful to assess potential loss of semi-volatile components.

I have a few remaining minor points, which I shall outline more briefly below.

Figure 1 presents the averaged particle number and mass distributions during a fog event. I assume these have been averaged over some time period corresponding to the fog event. The authors should indicate (1) whether the data have been averaged and (2) over what time period.

Some improvements could be made to the Methods section. Near the end of the paper, line 220 gives typical flow rate and power requirements for the self-developed and Palas drying systems. I think these would be better placed earlier, in the Methods section (say around line 125). Please also give the dimensions of the Palas' heated inlet for comparison. The authors should specify the orientation of the R1's inlet/dryer- is it pointed upwards as pictured in Figure 3A (similarly to the Palas inlet)? Within Methods, the authors say they used "sodium chloride, potassium chloride, ammonium sulphate, and ammonium nitrate" (line 111) with the aerosol generator, then go on to simply say how they used ammonium sulfate on lines 192/193. The information on lines 192/193 would be best moved to Methods, and I am not sure why these other salts have been included with no further discussion.

There are a few remaining points to clarify within the results sections. On line 158, the authors state the humidifier was turned off after reaching 300 $\mu gm^{-3}$. Given the differing RH sensitivity of the three instruments, the authors should state which of the three instruments was used to determine when this threshold had been reached.

Wet towels were added before switching the humidifier off in experiment 1 (fig 4A) but after in experiment 2 (fig 4B). It is not clear to me why this difference is present. Additionally, the analysis from line 157 would read more clearly if the authors actually name (before further discussion) which of the two experiments is being described (IADS set to automatic or 70°C). The authors simply use "Fig. 4a", "Fig. 4b" to indicate which experiment they are referring to, but I think this could be clearer.

Please give errors for the values of $\eta_r$ and $\eta_s$. Additionally, for the experiment with IADS at 70°C, only $\eta_s$ is given, please also specify $\eta_r$. The Palas settings have changed so it would be informative to compare the $\eta_r$ values from each experiment.

Do the authors know the typical temperature of the IADS during the automated setting experiment? It would be informative to compare this to the 70°C set temperature in the latter experiment.

In line 196, the authors discuss the lack of a "sudden decrease" in the Palas $PM_{2.5}$. I think it would be worth recognising that there is still a decrease, just slower than the R1 for the reasons discussed in the paper. The fact that the Palas $PM_{2.5}$ is less than the R1 without any dryer shows that some sample drying must be occurring, as the pre-experiment calibration should have largely removed other sources of disagreement.

In line 214, the author suggest the drops and jumps in $PM_{2.5}$ measured by the OPC R1 with dryer may be due to loss of semi-volatile components during heating. However the sample is ammonium sulfate (non-volatile), so where have the semi-volatile components come from? If unfiltered air has been used with the aerosol generator, this should be specified, particularly as this could also limit the extent of water uptake by the ammonium sulfate.

Finally, a few suggestions regarding the wording/readability in certain places.
- Line 98: "made from greenhouse glass"
- Line 106: "consisting of a"
- Line 124: "brass tube 50 cm in length"
- Line 200: "much more slowly"
- Line 203: "the decrease could have other causes"

Overall, good work! I found it very interesting to read.

References

Di Antonio, A. (2020). Development of novel methodologies for utilising low-cost sensors for ambient Particulate Matter measurement (Doctoral thesis).

---

## Author Comment (AC1)

**Response to interactive comments from Referee #1**

Thank you for the time you put into reviewing our manuscript and the helpful feedback. Please see our following responses and proposed changes to the original manuscript, which we believe, help to improve this paper and increase its impact. Below the comments from Referee #1 are given in black. Our responses to the comments are shown in blue. Text added or changed in the manuscript is marked in italics.

This work evaluates the performance of a low-cost thermal dryer for the OPC-R1, in comparison to an OPC-R1 without a dryer and the Palas Fidas 200, which has an automated drying system. Two separate scenarios which can lead to erroneously high PM readings at elevated relative humidity (RH) were investigated in a laboratory setting. (1) Fog conditions, where water droplets suspended in the air are incorrectly detected as particles, and (2) hygroscopic growth of particles, leading to elevated size and mass measurement relative to a measurement at lower RH. To evaluate the performance of the R1 with dryer against the other two instruments, both the time series and "drying efficiency" are presented. The authors conclude that the low-cost dryer was able to mitigate positive artifacts due to the two effects outlined above, but raise concerns over the comparability of their "fog" conditions to the real-world scenario. They state the need for further work to characterise the temperature profile and optimise the power requirements of their drying system.

This work is a useful addition to the literature on low-cost particulate matter sensors. The literature review is thorough and highlights that drying systems are both under-studied and potentially simpler to implement than postcalibration of measurements for RH. The two scenarios being investigated are clearly distinguished and explained.

I would like to highlight two main aspects of the paper which I think should be clarified. I think this is good work; my concerns relate to how the results are presented. I think the results could be more clearly placed within the context of standard reference measurements of PM, and certain aspects of the method are unclear.

We thank the reviewer for the detailed and insightful review and for the positive statement about our work. In the revised version, we have included a new section comparing the OPC-R1 sensor with low-cost dryer with gravimetric measurements and a "reference-equivalent" instrument during field measurements. We believe this new section places our results within the context of standard reference measurements of PM.

It is not clear to me what the authors are trying to achieve through use of their low-cost dryer. Multiple comparisons are made which all suggest a different target. Are they trying to (1) completely remove all water, (2) produce "reference-equivalent" PM readings to supplement regulatory monitors, or (3) simply reduce the positive artifact due to the presence of water? I think this needs to be stated from the outset as it affects which evaluation method is most appropriate.

In order to clarify the aim of the investigation, we have added the following text:

*"The aim of this study was to evaluate a prototype of a low-cost dryer built for a low-cost OPC under two different scenarios, namely fog events and hygroscopic growth, to reduce the influence of RH on the PM readings, i.e. to obtain "reference-equivalent" PM readings. For that purpose, experiments simulating both scenarios were performed under laboratory conditions and we quantified the effect of the dryer compared to a "reference-equivalent" monitor and to a low-cost OPC without a dryer. Additionally, two field campaigns were carried out with the aim of testing the prototype under real atmospheric conditions. In phase I, measurements with the gravimetric reference method, a continuous "reference-equivalent" monitor, and an OPC with dryer were performed in an urban background with daily averages of RH between 70 – 90 %. In phase II, measurements during a fog event with hourly averages of 100 % RH were carried out and the results of the OPC with dryer were compared to a continuous "reference-equivalent" monitor and to a sensor without dryer. Moreover, it was also evaluated whether the use of the low-cost dryer would allow a sensor calibration using exclusively a univariate linear regression (ULR) against gravimetric measurements, without the need of extra variables like RH."*

Comparing the R1 + dryer to the R1 without a dryer proves only aim (3), that water has been removed by heating. The reading is closer to dry particles, but we don't know how close. Comparing the R1 to the Palas may prove (1) or (2). My uncertainty here is due to the lack of information on how the Palas PM measurements were processed. The Palas can output (1) raw particle number size distribution, (2) PM "classic" and (3) PM "ambient". The authors could have obtained PM mass from (1) by applying their own density and penetration

factors, (2) is calculated by the Palas using a default assumed density $\rho = 1.3$ g/cm$^{-3}$, whilst (3) has an additional empirical correction factor applied to make the readings "reference-equivalent" (Di Antonio 2020).

We have added more information about the method used to obtain the mass concentration from the reference instrument:

"*The mass concentrations were directly obtained from the instrument using the "PM-Ambient" algorithms provided by the manufacturer.*"

Firstly, I would be unable to reproduce these findings without knowing which of the three methods the authors have used.

We have also added the following text in the conclusions for further clarification:

"*The investigated dryer has been proved to be very effective on reducing the water content of hygroscopic particles or fog. The results also indicate that our prototype dries the particles more than a reference-grade instrument, which suggest that this design of low-cost dryer is over-dimensioned in terms of heating power to have "reference-equivalent" PM readings.*"

Secondly, if the PM "ambient" readings have been used, I think it is worth pointing out that (under default IADS settings) the Palas is not trying to fully dry the particles because it is aiming to be reference-equivalent. Hence this is relevant to aim (2). The EU reference standard specifies conditioning for 24-hr at 20°C, 50% RH (see CEN standard 12341), ie not completely dry. Given this, the Palas can be expected to potentially measure PM which still contains bound water and not remove all fog droplets as it may not sufficiently lower the RH.

We have added the following text:

"*This result was expected, as the Fidas® 200 under default settings does not aim to completely dry the sampled air but seeks to meet the requirements for FEM instruments as set in the EU directive 2008/50/EC. These requirements are met when the PM readings of the FEM instrument correspond to the values of the measured PM filters of the standard gravimetric analysis after being pre-conditioned at 19 to 21 °C and 45 to 50 % RH for at least 48 h (EN 12341).*"

Thirdly, when the authors manually set the IADS to 70°C, they were overriding these reference-equivalent settings and were likely removing nearly all the water. This comparison is most appropriate to aim (1).

The following sentence has been added:

"*In order to determine an "apparent temperature" of the low-cost dryer, experiments in expert mode varying the temperature of the IADS were performed and the closest result is presented in Fig. 4b, in which the IADS was set using the expert mode at 70 °C. This "apparent temperature" is not the real temperature of the dryer as it was designed to keep a constant heat flux (through electric heating) and therefore the dryer has a temperature profile that varies through the length. More information about the air temperature inside the dryer has been summarized in section 3.1.3.*"

A short explanation of what each of the authors' comparisons (R1 without dryer, Palas automated settings, Palas at 70°C) actually means in the context of how much of the water is being removed and the relation to regulatory PM monitoring would make this paper more impactful.

The following sentence has been added:

"*The comparison with the reference instrument running in automatic mode shows how close the OPC with dryer is at getting "reference-equivalent" PM readings whereas comparing it with an OPC without dryer helps to quantify the amount of water that the dryer can actually remove.*"

A more precise description of how the Palas PM values were obtained is essential to make this work replicable by other authors. There should also be a more detailed description of how the R1 PM values were derived- were they the firmware-output values and what density and refractive index have been used? I would also point out that some discrepancy between R1 and Palas PM readings is to be expected because the R1 has a larger minimum detectable diameter.

Apart from the new text which describes how the mass concentration readings are calculated by the Fidas® 200 (see above), a new table collecting all the necessary technical specifications of the Fidas® 200 and the OPC-R1 (including density and refractive index as stated by the reviewer) has been added.

Moreover, the following text has been added to give more information about the derivation of the PM values for OPC-R1:

*"The mass concentrations were directly obtained from the PM outputs of the sensor."*

My second point of feedback is that the meaning of the $T_{OPC}$ quantity should be clarified. In particular, the authors should specify what this temperature corresponds to- is the sensor situated (1) on the OPC circuit board, or (2) within the sample air flow? The authors should then be clear about what they are preventing from overheating by including an upper temperature limit.

The following text has been added to specify where the temperature sensor is:

*"The temperature sensor is located in the OPC circuit board."*

In order to clarify what we mean with "to avoid overheating" we have added a new sentence,

*"Previous experiments showed that the OPC-R1 switches automatically off if $T_{OPC}$ reaches 44 °C so we selected an upper limit for $T_{OPC}$ of 35 °C."*

and rewritten the following sentence:

*"If $T_{OPC}$ is equal to or more than 35 °C the dryer switches off and starts cooling down to avoid overheating the sensor."*

I suspect case (1) applies, from my own work with the OPC-N3, a similar model of OPC. This makes TOPC the temperature of the OPC box/components rather than the sample. Therefore the upper temperature limit of 35 °C would be preventing the OPC from overheating and is not an appropriate control measure to prevent excessive sample heating.

The reviewer is right. In order to state this problem in the manuscript we have added the following text:

*"During the design phase, adding a temperature and a RH sensor at the end of the dryer was considered, but it was discarded as it would have affected the particle flow. The main advantage of using $T_{OPC}$ is the fact that it forms part of the OPC-R1, i.e. the $T_{OPC}$ data is part of the output of the sensor. However, the disadvantage is that using $T_{OPC}$ does not prevent sample overheating."*

As the authors state in the penultimate line, it is a shame that no information has been provided on the temperature (and RH) that the dryer conditions the sample to. This would be an important parameter in assessing which of the above three aims the dryer is best suited to. From the fact that the IADS had to be set to 70 °C to achieve comparable PM readings, I suspect that the sample is being heated much higher than the 35 °C limit on TOPC. This information would also be useful to assess potential loss of semi-volatile components.

We have added a new section 3.1.3 "Study on the drying temperature" containing the following text:

*"To get more information about the temperature profile inside the dryer, experiments were performed in the laboratory where the temperature of the air flowing inside the dryer was measured. The experiments showed that the maximum wall temperature is reached at 40 cm (Fig. S6). In the last centimeters the air is cooled down before the sensor inlet due to the lack of heated wire (the last 2.5 cm were left wire-free for ease of handling). It was observed that at 40 cm the air is heated up to approx. $65.9 \pm 0.5$ °C. This is in agreement with the experiments which show that the sensor with low-cost dryer behaves similar to reference instrument if the IADS is heated at 70 °C. As the thermocouple influences the air flow, the measured temperature may have some bias, but it is clear that it is higher than 40 °C, which is the maximum temperature recommended by the WMO/GAW guidelines for ambient air monitoring. Moreover, it was observed that the $T_{OPC}$ is usually 10 – 13 °C higher than the ambient temperature, which means that the dryer may not start heating when the ambient temperature is higher than 22 – 25 °C, as the $T_{OPC}$ could be already higher than the temperature limit set for the dryer (35 °C). This problem could be solved by changing the upper limit temperature loop in the Arduino code. However, this change also increases the maximum air temperature in the dryer, which is already too high for producing "reference-equivalent" PM readings. Therefore, we recommend that new versions of the low-cost dryer should*

*focus on the control of the RH in the sample flow, as the $T_{OPC}$ value is highly dependent on the ambient air temperature."*

I have a few remaining minor points, which I shall outline more briefly below.

Figure 1 presents the averaged particle number and mass distributions during a fog event. I assume these have been averaged over some time period corresponding to the fog event. The authors should indicate (1) whether the data have been averaged and (2) over what time period.

In order to make it clearer, we have changed the x axis in Fig. 1a so that it only covers the range of time that has been used to calculate the average of the size distributions in Fig. 1b and 1c.

We have also added some clarifications in the text as follows:

*"An example can be seen in Fig. 1a, where the one-minute average PM concentration…"*

*"The number of particles and the normalized mass concentration are averages over the time from 9:00 to 9:11h."*

Some improvements could be made to the Methods section. Near the end of the paper, line 220 gives typical flow rate and power requirements for the self-developed and Palas drying systems. I think these would be better placed earlier, in the Methods section (say around line 125). Please also give the dimensions of the Palas' heated inlet for comparison. The authors should specify the orientation of the R1's inlet/dryer- is it pointed upwards as pictured in Figure 3A (similarly to the Palas inlet)? Within Methods, the authors say they used "sodium chloride, potassium chloride, ammonium sulphate, and ammonium nitrate" (line 111) with the aerosol generator, then go on to simply say how they used ammonium sulfate on lines 192/193. The information on lines 192/193 would be best moved to Methods, and I am not sure why these other salts have been included with no further discussion.

As stated above, we have included a new table with the technical specifications of the Fidas® 200 and the OPC-R1 in the Methods section. The dimensions of the Palas inlet has been included in the text:

*"It is 1.2 m long and has an inner and an outer diameter of 12.7 and 48 mm, respectively."*

We have now pointed out the orientation of the low-cost dryer:

*"As it is shown in Fig. 2, the dryer is placed in a vertical position to minimize particle losses."*

The information in lines 192/193 has been moved to the Methods section.

Regarding the other salts, it was not included because the same conclusion is reached with all of them, the low-cost dryer always dries more than the IADS in automatic mode. We have now added two more interesting experiments in the supplemental material: one with pure ammonium nitrate and one with a mixture of the salts, and it has been indicated in the manuscript with the following text:

*"Experiments were carried out with different aerosols (($NH_4$)$_2SO_4$, $NH_4NO_3$, KCl and NaCl) and different IADS settings (automatic mode, IADS off (min. 20 °C), 35 °C, 50 °C, and 65 °C)."*

*"For this experiment, the Fidas® 200 ran in automatic mode. Experiments with $NH_4NO_3$ and the mixture of the salts can be seen in Fig. S3 and S4 of the supplemental material, respectively. The DRH and ERH of ($NH_4$)$_2SO_4$ as well as the other tested salts are indicated in Table S2 in the supplemental material."*

There are a few remaining points to clarify within the results sections. On line 158, the authors state the humidifier was turned off after reaching 300 μg m$^{-3}$. Given the differing RH sensitivity of the three instruments, the authors should state which of the three instruments was used to determine when this threshold had been reached.

The following clarification has been added:

*"After the reference instrument reached a PM2.5 mass concentration of 300 μg m$^{-3}$,…."*

Wet towels were added before switching the humidifier off in experiment 1 (fig 4A) but after in experiment 2 (fig 4B). It is not clear to me why this difference is present. Additionally, the analysis from line 157 would read more clearly if the authors actually name (before further discussion) which of the two experiments is being described (IADS set to automatic or 70°C). The authors simply use "Fig. 4a", "Fig. 4b" to indicate which experiment they are referring to, but I think this could be clearer.

During the evaluation of the experiment with Fidas® 200 in automatic mode we observed that the RH was decreasing after switching off the humidifier. In order to avoid that decrease, we decided to introduce the wet towels before.

We have rewritten the following sentences to clarify which of the experiments we are referring to:

*"…, in Fig. 4a, the IADS of the reference instrument was kept in automatic mode, i.e. the default settings under which the instrument works during field measurements, whereas in Fig. 4b, it was set at 70 °C using the expert mode."*

*"As shown in Fig. 4a during the experiment with the IADS in automatic mode,…"*

Please give errors for the values of $\eta_r$ and $\eta_s$. Additionally, for the experiment with IADS at 70°C, only $\eta_s$ is given, please also specify $\eta_r$. The Palas settings have changed so it would be informative to compare the $\eta_r$ values from each experiment.

We have added the errors of $\eta_r$ and $\eta_s$ calculated with the standard deviation. The second part of this comment is really good because it made us re-evaluate the meaning of $\eta_r$. As it was defined in Equation 1, it was difficult to understand the meaning (this is why it was not included in that experiment). We have now made a simple change, that is, instead of calculating the term "$1 - \frac{PM2.5_{d,i}}{PM2.5_{r,i}}$", we use only $\frac{PM2.5_{d,i}}{PM2.5_{r,i}}$, so the meaning is more comprehensible as it is explained now in the text as follows:

*"Each drying efficiency provides different information. The $\eta_r$ gives an idea about how close the average PM2.5 readings are between the reference instrument and the sensor with low-cost dryer. In other words, the higher the $\eta_r$ the closer the PM2.5 to "reference-equivalent" PM2.5 readings. The $\eta_s$, in contrast, helps to estimate the actual drying capacity of the low-cost dryer. In the experiments with the air humidifier, it is possible to estimate with $\eta_s$ the ability of the low-cost dryer of removing water from the sample flow. In the case of the experiments with hygroscopic salts, $\eta_s$ estimates the ability of the low-cost dryer to avoid hygroscopic growth."*

Do the authors know the typical temperature of the IADS during the automated setting experiment? It would be informative to compare this to the 70°C set temperature in the latter experiment.

Yes, we have this information and we have added three figures (one for each experiment) with the IADS temperatures and heating power in a new supplement (Fig. S1, S2, S5 and S9).

In line 196, the authors discuss the lack of a "sudden decrease" in the Palas PM2.5. I think it would be worth recognising that there is still a decrease, just slower than the R1 for the reasons discussed in the paper. The fact that the Palas PM2.5 is less than the R1 without any dryer shows that some sample drying must be occurring, as the pre-experiment calibration should have largely removed other sources of disagreement.

The reviewer is right. We have changed the sentence to:

*This decrease is also observed with the reference instrument, but at lower pace.*

In line 214, the author suggest the drops and jumps in PM2.5 measured by the OPC R1 with dryer may be due to loss of semi-volatile components during heating. However the sample is ammonium sulfate (non-volatile), so where have the semi-volatile components come from? If unfiltered air has been used with the aerosol generator, this should be specified, particularly as this could also limit the extent of water uptake by the ammonium sulfate.

Actually, the loss of semi-volatile components due to the excess of heating was meant for field measurements. Consequently, it is not related to the experiment. As the reviewer rightly stated, the ammonium sulfate particles used are non-volatile. We have moved the text into a new Discussion section so it is clear that it is not related to the experiments and the general thoughts and discussion start.

Finally, a few suggestions regarding the wording/readability in certain places.

• Line 98: "made from greenhouse glass"

We have changed this.

• Line 106: "consisting of a"

We have changed this.

• Line 124: "brass tube 50 cm in length"

We have changed this.

• Line 200: "much more slowly"

We have changed this.

• Line 203: "the decrease could have other causes"

We have changed this.

Overall, good work! I found it very interesting to read.

Thank you so much!

**References**

Di Antonio, A. (2020). Development of novel methodologies for utilising low-cost sensors for ambient Particulate Matter measurement (Doctoral thesis).

Thank you for this reference. We were not aware that Di Antonio also tested a thermal dryer for OPC-N2. We have included this work in the references.

---

## Author Comment (AC2)

**Response to interactive comments from Referee #2**

Thank you for the time you put into reviewing our manuscript and the helpful feedback. Please find below our responses and proposed changes to the original manuscript, which improve the manuscript. Below the comments from Referee #2 are given in black. Our responses to the comments are shown in blue. Text added or changed in the manuscript is marked in italics.

The manuscript presents an interesting application of a heater/dryer to lessen the impact of high humidity/fog events on OPC performance. The experiments are carefully executed and clearly described. There are no major problems with the text or content, other than a general lack of statistical analysis (correlation, deviation, statistical significance etc).

Thank you for the assessment of our work. In order to cover the lack of statistical analysis we have added the errors calculated with the standard deviation of the drying efficiencies. Moreover, we have included field measurements to compare the OPC-R1 with low-cost dryer to gravimetric analysis and a "reference-equivalent" monitor under real conditions. The results include three new tables with a summary of statistics (SD, MAE, RMSE, MBE, slope, offset, R² and Pearson coefficient).

However, I remain unconvinced of the arguments given that this hardware solution to the problem is merited over software solutions, which are presented as complex and difficult. It is acknowledged in the paper that calibration of the OPC is still required. Software solutions are theoretically capable of addressing several limitations of OPCs, including high humidity, variation in composition, size distribution etc. This hardware solution only addresses one, so I would argue that software solutions of a similar complexity will still be required even in units with a dryer fitted.

As we have described in the introduction, all currently available solutions to avoid the negative effect of high RH on sensor readings, hardware as well as software ones, have advantages and disadvantages. As the reviewer states, software solutions are theoretically capable of addressing several limitations of OPCs, but they do have limitations too. These limitations have already been described in the original manuscript in lines 67 to 78.

We can argue that neither the low-cost dryer nor the software solution will make the sensor to be exempted of calibration. Calibration is always needed, even for traditional air quality monitors and reference instruments. The advantage of using a dryer would be to calibrate/correct the sensors signal against a reference instrument with a univariate linear regression without the additional need of a software. This is in our opinion a more citizen-friendly/Do-It-Yourself method than a software solution.

In order to address these points in the manuscript, we have added the following text:

*"It is worth noting that even though drying the air could arise questions about how the temperature affects the physico-chemical properties of the particulate matter, existing software solutions are not problem-free, as they may fail when changes in the particle chemical composition occurs. Moreover, a software solution that helps to minimize the effect of the hygroscopic growth and fog events in the mass concentrations has not been reported in the literature. In general, the effect of fog in the mass concentrations of sensors has scarcely been addressed in the literature."*

Furthermore, dryers introduce problems of their own, notably the driving off of semivolatile PM. The authors acknowledge this but do not propose a solution, just advising a compromise. This compromise may negate the benefit of the hardware solution over existing software solutions. These two facts together suggest that a dryer is an incomplete solution that will increase hardware cost and complexity but not necessarily decrease the need for software correction except, perhaps, in very specific high frequency foggy geography. The authors are welcome to attempt to address these points to make a more convincing argument as to the utility of a dryer.

We admit that our dryer may introduce problems of its own but, as explained in the manuscript, software solutions are not the "holy grail" either. Our prototype may not be the perfect solution but can be a source of inspiration for other researchers to improve it, or it could be combined with software solutions.

We have added the following text:

*"It is clear that PM sensors have come to the air quality monitoring market to stay, and that (i) all new approaches (hardware, software or hybrid-solutions) which aim to improve the accuracy of PM sensors and (ii)*

*evaluations on how they behave when the environmental conditions change due to e.g. fog events, long-range transport, or a change in sensor location, are welcome to be addressed in future research."*

Other points:

Abstract - The abstract does not describe the results well - lower PM2.5 concentration in itself does not equate to improved sensor performance or accuracy. Statistics should relate to statistical comparison with reference method results.

*We have included a statistical analysis comparing the results of the low-cost dryer with the standard method. Moreover, we have substantially modified the abstract to read as follows:*

*"The use of low-cost sensors for air quality measurements has become very popular in the last decades. Due to the detrimental effects of particulate matter (PM) on human health, PM sensors like photometers and optical particle counters (OPC) are widespread and have been widely investigated. The negative effects of high relative humidity (RH) and fog events in the mass concentration readings of these types of sensors are well documented. In the literature, different solutions to these problems - like correction models based on the Köhler theory or machine learning algorithms - have been applied. In this work, an air pre-conditioning method based on a low-cost, thermal dryer for a low-cost OPC is presented. This study was done in two parts. The first part of the study was conducted in laboratory to test the low-cost dryer under two different scenarios. In one scenario, the drying efficiency of the low-cost dryer was investigated in the presence of fog. In the second scenario, experiments with hygroscopic aerosols were done to determine to which extent the low-cost dryer reverts the growth of hygroscopic particles. In the second part of the study, the PM10 and PM2.5 mass concentrations of an OPC with dryer were compared to gravimetric measurements and a continuous Federal Equivalent Method (FEM) instrument in the field. The feasibility of using univariate linear regression (ULR) to correct the PM data of an OPC with dryer during field measurement was also evaluated. Finally, comparison measurements between an OPC with dryer, an OPC without dryer and a FEM instrument during a real fog event are also presented. The laboratory results show that the sensor with the low-cost dryer at its inlet measured an average of 64 % and 59 % less PM2.5 concentration compared to a sensor without the low-cost dryer during the experiments with fog and with hygroscopic particles, respectively. The outcomes of the PM2.5 concentrations of the low-cost sensor with dryer in laboratory conditions reveals, however, an excess of heating compared to the FEM instrument. This excess of heating is also demonstrated in a more in-depth study on the temperature profile inside the dryer. The correction of the PM10 concentrations of the sensor with dryer during field measurements by using ULR showed a reduction of the maximum absolute error (MAE) from 4.3 $\mu g\ m^{-3}$ (raw data) to 2.4 $\mu g\ m^{-3}$ (after correction). The results for PM2.5 make evident an increase in the MAE after correction: from 1.9 $\mu g\ m^{-3}$ in the raw data to 3.2 $\mu g\ m^{-3}$. In light of these results, a low-cost, thermal dryer could be a cost-effective add-on that could revert the effect of the hygroscopic growth and the fog in the PM readings. However, special care is needed when designing a low-cost dryer for a PM sensor to produce FEM similar PM readings, as high temperatures may irreversibly change the sampled air by evaporating the most volatile particulate species and thus deliver underestimated PM readings. New versions of a low-cost dryer aiming at FEM measurements should focus on maintaining the RH at the sensor inlet at 50 %, and avoid reaching temperatures higher than 40 °C in the drying system. Finally, we believe that low-cost dryers have a very promising future for the application of sensors in citizen science, in sensor networks for supplemental monitoring, and for epidemiological studies."*

Introduction - the example shown focuses the discussion on PM10, yet the publication focuses on PM2.5. Rewrite text with focus on PM2.5 as OPC have other issues not addressed when sampling PM10.

*In the new results for field measurements, we have included PM10 as well as PM2.5. Moreover, we believe it is important to clarify the particle size distribution of fog even though in the laboratory experiments only the PM2.5 are presented. Nevertheless, we have added a new sentence in the introduction.*

*"PM2.5 and PM1 are in the range of $10^2 – 10^3$ and $10^1 – 10^2$ $\mu g\ m^{-3}$, respectively."*

---

## Author Comment (AC3)

**Response to interactive comments from Referee #3**

Thank you for the time you put into reviewing our manuscript and the helpful feedback. Please find in the following our responses and proposed changes to the original manuscript, which improve the manuscript. Below the comments from Referee #3 are given in black. Our responses to the comments are shown in blue. Text added or changed in the manuscript is marked in italics.

The paper presents the application of a heated inlet to mitigate against the influence of hygroscopic growth of particles on the imputed particle size distribution and mass loading by an OPC, in this case the 'lower-cost' OPC-R1. The paper is well written and the procedures are mostly clear, though considering that this seems to be an 'open-sourced' project, it would be helpful to have more specific information on the construction of the heater – more information on electronics is included in the linked Github page, but not sufficient to recreate the design without significant leaps. However, my main concern is that it the interpretation of results is such that it is difficult to know whether such a device will actually improve measurements by low-cost sensors, or introduce other, possibly more difficult to correct errors. Additional experiments that seek to separate 'drying' from 'particle evaporation' (by varying particle volatility or treatment mechanisms) could do this, but are a substantial extension beyond what is included here.

We have updated the document for Zenodo/Github with more information about the dryer construction and the electronics.

Regarding the interpretation of the results, we have now included results of field campaigns where the PM concentrations of the low-cost dryer are compared to gravimetric analysis and "reference-equivalent" monitors during "real" conditions of hygroscopic growth and fog.

In general, I share similar concerns to the previous two reviewers, and offer some specific points that should be addressed below. I am in particular concerned about the influence of and lack of 'control' on the temperature of the / sample and the fact that it may be as high as 70 degrees C. We don't really know what the temperature it is, but it appears it may be high as this is when the most consistent behavior with the reference instrument is observed (when the Palas was operated at a fixed temperature). The paper (rightly) includes multiple allusions to the potential for loss of semivolatile material, and this is indeed a major concern for application of this system for ambient aerosol, especially in urban areas where most lowcost sensors are deployed. Heated tubes of this type are used as 'thermodenuders' to remove semivolatile organics, and previous deployments in urban areas find that nearly 50% of organic aerosol will be removed by heating to 70 C (Paciga et al. 2016), though with a longer residence time (I calculate around 8 seconds for your heater geometry vs. 50 seconds for the Paciga et al system). If organic aerosol or ammonium nitrate are substantial components of the sampled aerosol, this heater will remove much of this material and bias any measurements low. The removal of water and semivolatile removal components could have been isolated by running experiments with diffusion or Nafion drivers in parallel with the heated tube. However, in the absence of such data and especially in the absence of information about the actual temperature of heating, it is (and will be) hard to interpret what the heater is actually doing to the aerosol. This is especially tricky because much of fine aerosol number and mass will be in the submicron range, and evaporation will push the entire size distribution out of the range of diameters the OPC(s) can detect.

We have added a new section 3.1.3 "Study on the drying temperature" where we include more information about the air temperature in the dryer. The drying process in our prototype is complex, due to the discontinuity of the dryer but, it is actually as the reviewer assumed, that temperatures higher than 40 °C are reached when the dryer is on and that therefore part of the semi-volatiles is lost. How much time the dryer is on depends on the ambient temperature which makes difficult to control the temperature. In order to address this problem the following text has been included:

*"To get more information about the temperature profile inside the dryer, experiments were performed in the laboratory where the temperature of the air flowing inside the dryer was measured. The experiments showed that the maximum wall temperature is reached at 40 cm (Fig. S6). In the last centimeters the air is cooled down before the sensor inlet due to the lack of heated wire (the last 2.5 cm were left wire-free for ease of handling). It was observed that at 40 cm the air is heated up to approx. $65.9 \pm 0.5$ °C. This is in agreement with the experiments which show that the sensor with low-cost dryer behaves similar to reference instrument if the IADS is heated at 70 °C. As the thermocouple influences the air flow, the measured temperature may have some bias, but it is clear that it is higher than 40 °C, which is the maximum temperature recommended by the WMO/GAW guidelines for ambient air monitoring. Moreover, it was observed that the $T_{OPC}$ is usually $10 - 13$ °C higher than*

*the ambient temperature, which means that the dryer may not start heating when the ambient temperature is higher than 22 – 25 °C, as the $T_{OPC}$ could be already higher than the temperature limit set for the dryer (35 °C). This problem could be solved by changing the upper limit temperature loop in the Arduino code. However, this change also increases the maximum air temperature in the dryer, which is already too high for producing "reference-equivalent" PM readings. Therefore, we recommend that new versions of the low-cost dryer should focus on the control of the RH in the sample flow, as the $T_{OPC}$ value is highly dependent on the ambient air temperature."*

There is a strong basis for the limitation of inlet temperature to 40 C in 'reference' instruments (as noted by WMO/GAW guidelines) due to the influences discussed above. For example, early studies with the 'TEOM' found a strong and variable bias due to loss of semivolatiles due to heating (typically to 50 C) (Allen et al. 1997; Charron et al. 2004) and later versions did away with this heating.

As mentioned in the previous comment, we have addressed this problem in a new section (see comment above).

We have added a new Discussion section in which this topic is addressed in the following sentence:

*"Additionally, a temperature limit of 40 °C should be introduced, as recommended by the WMO/GAW guidelines."*

The 'fog' measurements, don't appear to really be fog, but are likely residual contaminants from the humidifier. This is evident by the small size (as noted by the authors) relative to actual fog droplet. There's a wide literature on this, going back decades (e.g. (Rodes et al. 1990)). Therefore, while the application of this heater systems for aerosol measurements in foggy environments may be a goal, it doesn't appear to be one tested here. Rather, these are similar experiments to the others shown, but with aerosols of unknown composition.

We were aware that the mineral composition in the water could have an effect in the experiments. Consequently, we selected the humidifier U350 that integrates a filter unit (250 AQUA PRO). Moreover, test experiments were performed using deionized water without a significant difference in the results. We have also double-checked with recent literature and the particle distribution we obtained corresponds to what other researchers have obtained for deionized water using ultrasonic humidifiers (Sain et al. 2018). We have added the following in the manuscript:

*"This model of humidifier integrates a filter unit (250 AQUA PRO) that allows the generation of pure water droplets."*

The results in Figure 6 may be helpful to separate 'drying' from 'evaporating' because they are with a non-volatile aerosol at known RH. However, they are still difficult to interpret. I expect this is because there are interactions between different drying conditions (the low-cost dryer is probably 'over drying' relative to reference conditions) and the size cutoff of the two OPCs (the R1 may be 'missing' a substantial amount of material between 0.18 and 0.3 microns).

The reviewer is right, the OPC with dryer is over drying relative to the Fidas® 200. We believe the new section "Study on the drying temperature" help to interpret the drying process in the low-cost sensor. We have also added the following sentence:

*"It should be also highlighted that approx. 80 % of the particles seen by the Fidas® 200 have a mean diameter from 0.17 to 0.35 µm, which means that the OPCs are not detecting a substantial amount of material."*

The paper alludes to testing various inorganic aerosols (Line 111) but these data aren't included. In particular, tests with ammonium nitrate would likely highlight the influence of heating on semivolatile material. This additional data should be included/discussed.

We have included one experiment with ammonium nitrate as well as an experiment with a mixture of the aerosols in the supplemental material.

Specific points

L15 – Not clear that comparing average PM2.5 concentrations are appropriate comparisons. Other distribution parameters and comparative statistics are helpful.

We have included more statistics for the field measurements and updated the full abstract to the following:

*"The use of low-cost sensors for air quality measurements has become very popular in the last decades. Due to the detrimental effects of particulate matter (PM) on human health, PM sensors like photometers and optical particle counters (OPC) are widespread and have been widely investigated. The negative effects of high relative humidity (RH) and fog events in the mass concentration readings of these types of sensors are well documented. In the literature, different solutions to these problems - like correction models based on the Köhler theory or machine learning algorithms - have been applied. In this work, an air pre-conditioning method based on a low-cost, thermal dryer for a low-cost OPC is presented. This study was done in two parts. The first part of the study was conducted in laboratory to test the low-cost dryer under two different scenarios. In one scenario, the drying efficiency of the low-cost dryer was investigated in the presence of fog. In the second scenario, experiments with hygroscopic aerosols were done to determine to which extent the low-cost dryer reverts the growth of hygroscopic particles. In the second part of the study, the PM10 and PM2.5 mass concentrations of an OPC with dryer were compared to gravimetric measurements and a continuous Federal Equivalent Method (FEM) instrument in the field. The feasibility of using univariate linear regression (ULR) to correct the PM data of an OPC with dryer during field measurement was also evaluated. Finally, comparison measurements between an OPC with dryer, an OPC without dryer and a FEM instrument during a real fog event are also presented. The laboratory results show that the sensor with the low-cost dryer at its inlet measured an average of 64 % and 59 % less PM2.5 concentration compared to a sensor without the low-cost dryer during the experiments with fog and with hygroscopic particles, respectively. The outcomes of the PM2.5 concentrations of the low-cost sensor with dryer in laboratory conditions reveals, however, an excess of heating compared to the FEM instrument. This excess of heating is also demonstrated in a more in-depth study on the temperature profile inside the dryer. The correction of the PM10 concentrations of the sensor with dryer during field measurements by using ULR showed a reduction of the maximum absolute error (MAE) from 4.3 µg m$^{-3}$ (raw data) to 2.4 µg m$^{-3}$ (after correction). The results for PM2.5 make evident an increase in the MAE after correction: from 1.9 µg m$^{-3}$ in the raw data to 3.2 µg m$^{-3}$. In light of these results, a low-cost, thermal dryer could be a cost-effective add-on that could revert the effect of the hygroscopic growth and the fog in the PM readings. However, special care is needed when designing a low-cost dryer for a PM sensor to produce FEM similar PM readings, as high temperatures may irreversibly change the sampled air by evaporating the most volatile particulate species and thus deliver underestimated PM readings. New versions of a low-cost dryer aiming at FEM measurements should focus on maintaining the RH at the sensor inlet at 50 %, and avoid reaching temperatures higher than 40 °C in the drying system. Finally, we believe that low-cost dryers have a very promising future for the application of sensors in citizen science, in sensor networks for supplemental monitoring, and for epidemiological studies."*

L18 – Here and elsewhere there is discussion of 'accuracy' of sensors, but nowhere is data from this arrangement compared to a 'true' reference measurements, and so this seems a bit of a bold claim.

We agree that the results are difficult to interpret in the context of standard reference measurements without comparing with the "true" reference. Therefore, we have added new results that includes a comparison of the OPC-R1 with gravimetric measurements during field deployment.

L31 – Accuracy is one limiting factor, but there are other key concerns: power consumption, durability, … . What are 'certain applications'?

We have changed it to the following text:

*"The accuracy needed for certain applications e.g. regulatory air quality monitoring or environmental epidemiology is at this moment one of the limiting factors for the use of low-cost sensors."*

Figure 1- These data traces appear to be smoothed. This is not appropriate for discrete data points.

Plotted data in Fig. 1 are now unsmoothed.

L76 – It's not clear what 'quality' means here. A key considering is that the variable space covered by training and deployment data sets coincide.

We have changed it to the following sentence:

*"However, they also have limitations such as the high dependency on the quality (accuracy of all input variables, outlier detection) and length of the training data and the extensive computational resources required."*

L138 – As noted above and by another referee, a lack of specificity about what exactly this temperature refers to is a key question. The key temperature for evaporation of semivolatiles will the air temperature within the heater, as semivolatile materials evaporated will be a function of this temperature.

As stated in a previous answer to a comment, we have included a new section about the air temperature inside the dryer and added some more information in the supplemental material.

Eq. 1 – This approach doesn't seem helpful, as it combines differences in instrument response (e.g. due to different size cuts and calibrations) with 'drying' (which is actually drying + evaporation of semivolatiles)

Thank you for the note. We have made a modification in Equation 1 that we believe helps to understand the meaning of the drying efficiency: instead of calculating the term "$1 - \frac{PM2.5_{d,i}}{PM2.5_{r,i}}$", only $\frac{PM2.5_{d,i}}{PM2.5_{r,i}}$ is now calculated. We have added also the following explanation:

*"Each drying efficiency provides different information. The $\eta_r$ gives an idea about how close the average PM2.5 readings are between the reference instrument and the sensor with low-cost dryer. In other words, the higher the $\eta_r$ the closer the PM2.5 to "reference-equivalent" PM2.5 readings. The $\eta_s$, in contrast, helps to estimate the actual drying capacity of the low-cost dryer. In the experiments with the air humidifier it is possible to estimate with $\eta_s$ the ability of the low-cost dryer of removing water from the sample flow. In the case of the experiments with hygroscopic salts, $\eta_s$ estimates the ability of the low-cost dryer to avoid hygroscopic growth."*

L149 – R2 isn't 'coefficient of correlation' – this also seems to be a minimal/insufficient way to compare these data sets under the 'best possible' conditions. What about slope/offset/bias?

We have changed it to coefficient of determination and added more statistics in Table 3, 4 and 5.

L159 – Given the size distribution measured, it is likely not sedimentation, but rather diffusive loss to chamber surfaces.

We have removed "the sedimentation curve started".

L161 – The different size cut of the two OPCs could be a factor here. A minor growth could push particles into the detection window of the Palas. So this is likely not more particles, but growth of the same ones.

That is exactly what we meant by "increase of the water droplets which were too small to be detected at lower RH". For better explanation, we have changed the sentence to the following:

*"...possibly due to the growth of water droplets which were below the detection limit of the instruments at lower RH."*

L164 – As noted by other reviewer, this is likely because the dryer is going well beyond 'reference' conditions, and probably evaporating some of the salts/organics in these particles.

We have added the following text:

*"This result was expected, as the Fidas® 200 under default settings does not aim to completely dry the sampled air but seeks to meet the requirements for FEM instruments as set in the EU directive 2008/50/EC. These requirements are met when the PM readings of the FEM instrument correspond to the values of the measured PM filters of the standard gravimetric analysis after being pre-conditioned at 19 to 21 °C and 45 to 50 % RH for at least 48 h (EN 12341)."*

L168 – As noted above, these are not fog droplets, but probably particles of residuals from humidifier.

As we have explained before, the experiments were done with fog droplets as the possible residuals were filtered by the humidifier.

L189 – Previous work in this area has determined that there is not an 'optimum' and recommends the 40 C upper limit. This may be an important need, but as indicated by other reviewer, it needs to be made clear that this is clearly beneficial relative to computational approaches.

We have included the following sentence:

*"One possible solution is introducing an adaptive heating to the dryer control to keep the RH of the air at the sensor inlet constant at 50 %. In such a case the temperature needed to maintain the RH of the air at 50 % could be adjusted so that higher temperatures than 40 °C would only be reached in during fog events, where the RH is close to 100% in order to be able to counter-react the effect of the fog in the PM readings. As can be seen in Fig. S8 in the supplemental material, the IADS also achieved temperatures higher than 40 °C during the real fog event."*

L203 – As noted above, sedimentation is likely not an important loss process for particles in this size range (all sub-micron)

We have changed it to the following:

*"...the decrease could have other causes, for instance, the sedimentation of the heavier particles or particle deposition onto the wall."*

L235 – Not clear whether there would be any benefit to drying flow to an electrochemical (or other gas) sensor, as they typically respond to both temperature and RH, and possibly to absolute humidity. Therefore, a 'dryer' that doesn't actually remove water won't necessarily help (and may make things more complicated when it comes to signal interpretation)

A dryer for electrochemical sensors has already been tested in our laboratory and the results can be read in Samad et al. 2020. We have added the following text:

*"Moreover, the design of the dryer can be easily adapted to other models or types of sensors, including, for instance, electrochemical sensors for gases as it has been tested in Samad et al. (2020)"*

References

Allen, G., C. Sioutas, P. Koutrakis, R. Reiss, F. W. Lurmann, and P. T. Roberts. 1997. "Evaluation of the TEOM(R) method for measurement of ambient particulate mass in urban areas." J. Air Waste Manag. Assoc., 47 (6): 682–689.

Charron, A., R. M. Harrison, S. Moorcroft, and J. Booker. 2004. "Quantitative interpretation of divergence between PM10 and PM2.5 mass measurement by TEOM and gravimetric (Partisol) instruments." Atmos Env. Atmos Env., 38 (3): 415–423.

Paciga, A., E. Karnezi, E. Kostenidou, L. Hildebrandt, M. Psichoudaki, G. J. Engelhart, B.-H. Lee, M. Crippa, A. S. H. Prévôt, U. Baltensperger, and S. N. Pandis. 2016. "Volatility of organic aerosol and its components in the megacity of Paris." Atmos Chem Phys, 16 (4): 2013–2023. https://doi.org/10.5194/acp-16-2013-2016.

Rodes, C., T. Smith, R. Crouse, and G. Ramachandran. 1990. "Measurements of the Size Distribution of Aerosols Produced by Ultrasonic Humidification." Aerosol Sci. Technol., 13 (2): 220–229. Taylor & Francis. https://doi.org/10.1080/02786829008959440.

Sain, A. E.; Zook, J.; Davy, B. M.; Marr, L. C.; Dietrich, A. M. (2018): Size and mineral composition of airborne particles generated by an ultrasonic humidifier. In Indoor Air 28 (1), pp. 80–88. DOI: 10.1111/ina.12414.

Samad, Abdul; Obando Nuñez, Daniel Ricardo; Solis Castillo, Grecia Carolina; Laquai, Bernd; Vogt, Ulrich (2020): Effect of Relative Humidity and Air Temperature on the Results Obtained from Low-Cost Gas Sensors for Ambient Air Quality Measurements. In Sensors (Basel) 20 (18). DOI: 10.3390/s20185175.

---

## Author Response (AR2)

**"Evaluation of a Low-Cost Dryer for a Low-Cost Optical Particle Counter"**

**Response to Report #1**

I commend the authors for the substantial improvements and clarifications they have made in this revision. I think the potential utility of their system is highlighted and the experiments and data are generally clearly articulated. Most of my concerns have been addressed, but there remain some issues… In my view, the limitations of the system as it was operated in this study are not clearly enough stated using the data they have and insights from other work. It is important to both show that this is a contribution but also be very clear about its limitations. The most important limitation in my mind is that it is clear that the aerosol is being heated far more than it should be, and that this is likely causing excess water, and perhaps more importantly, any semivolatile components (ammonium nitrate and organics, most notably) to evaporate. The revised manuscript makes it clear this is an issue, but several times calls it 'over drying' (it's not drying, but drying and evaporation) and does little to say why/by how much. For example, the supplement contains data (Fig S3) that demonstrate this issue (extensive evaporation of ammonium nitrate particles in the heater and in the IADS), but the manuscript simply mentions the figure (line 387) without discussing it and its implications. Fig S3 shows a ~5-fold under prediction with the dryer and ammonium nitrate (Fig S3), which will introduce huge bias if this or organics are dominant components of PM. In fact, in the phase I field measurements you specifically note that high concentrations of ammonium nitrate (which is hydroscopic, but also semivolatile) are expected. The degradation in PM2.5 performance with the heater is likely due to this. My feeling is that this issue needs to be addressed directly.

We thank the reviewer for his comments and the time spent in undergoing the review process. In the new version of the manuscript, we have discussed Fig. S3 and added its implications. The text is as follows:

*"An experiment with $NH_4NO_3$ particles is shown in Fig. S3, in which different IADS temperatures were manually set (20 °C, 35 °C, 50 °C and 65 °C). This experiment clearly shows the impact of temperature in the loss of semi-volatiles due to evaporation and, therefore, the detrimental effect that the use of high temperatures in a heated inlet can have on the mass concentration when species with high volatility like $NH_4NO_3$ are present in the sample. In this sense, the presented design of a low-cost dryer is behaving as a thermodenuder, i.e., a device that is used to study the volatility fraction of aerosol particles (Huffman et al., 2008). Studies using thermodenuders have shown that temperatures of 83 to 88 °C can cause 50 % of the organic aerosol mass to evaporate (Paciga et al., 2016). For the specific case of nitrate, much lower temperatures are needed to reduce the mass by 50 %, as it is shown in the results of Huffman et al. (2009) where 50 % of the nitrate during a field campaign was evaporated at 54 °C."*

Regarding the field measurements, we agree with the reviewer and think that the degradation of the PM2.5 performance during the field measurements is likely due to the evaporation of the semi-volatile species, as already stated in the previous version of the manuscript: *"… the PM2.5 is frequently underestimated. This underestimation occurs probably due to two reasons: (1) most of the semi-volatile organic compounds belong to the PM2.5 fraction and the dryer could be evaporating them."* Additionally, we have added the following sentence:

*"It is likely that reason (1) prevails over (2), as a significant amount of ammonium sulphate and, especially, ammonium nitrate, is expected in the PM2.5 fraction."*

In my first review I pointed out an example of the use of a heated tube as a thermodenuder, which is essentially what this is, and how it may influence aerosol mass transmitted to the sensor. Considering the high temperature (nearly 70 deg. C) reached in the heater, all results should be viewed in this context (how much bias may be introduced), rather than just including a caveat that this is an issue. However, no mention of this literature or the data that was collected showing this to be an issue is included. Another paper, which shows similar data for nitrate is (Huffman et al. 2008), or for lab-generated ammonium nitrate see Fig 4a here: (Huffman et al. 2009).

We have included a comparison of the low-cost dryer with a thermodenuder and the mentioned literature following the discussion of Fig. S3.

In a related issue, my suspicion is that the results in Fig. 4 a and 4b are being misinterpreted, which are as much showing evaporation of particles and not removal of water. The fact that the Fidas and the undried sensor in Fig. 4a seem to agree suggests that the particles formed during the humidification stage have not taken up excess water (otherwise you'd see a big difference) and what you're seeing when the heater turns at 65 minutes on is evaporation of semivolatile material in the particles. This is confirmed in 4b, where you see that heating to 70 deg. C, much more than needed to remove water, is greatly reducing the Fidas signal and making it align with your heated sensor data.

*After further inspection of the literature, we agree with the reviewer on the fact that the ultrasonic humidifier is also generating particles that have their origin in the minerals present in the tap water. These impurities are mainly calcium, magnesium and sodium among other compounds. That means that what we probably measured was the evaporation of water that condensed on the impurities, which had acted as condensation nuclei. In the laboratory experiments carried out by Lau et al. (2021), a diffusion dryer is used to dry the water droplets before sample collection which means that he was also concerned that the impurities were not fully dry and that a determined content of water remains absorbed in the residuals.*

*For clarification in the manuscript we have included the following sentences:*

*"As shown in Fig. 4a during the experiment with the IADS in automatic mode, once the air humidifier was on, tiny water droplets containing impurities were generated. The water droplets evaporated quickly and, as a consequence, the RH started to increase, leaving the solid impurities with associated water as suspended particles in the air."*

*"...a remarkable increase in the PM2.5 concentration was observed, possibly due to the water uptake of the impurities."*

*"Once the RH reached 65 %, the low-cost dryer of the OPC-R1 started heating and a pronounced decrease in the PM2.5 concentration is observed, probably due to not only the evaporation of the water but also the evaporation of semi-volatile species."*

As I noted in my first review, you are not measuring fog droplets in the lab experiments as claimed on lines 240 and 347 and elsewhere in Sec. 3.1.1, but measuring mineral residue with some associated water. Even deionized water has some residuals, and these are not 'filtered' by the atomizer. You are atomizing water that has some dissolved solids (not 'pure water'), and even very low concentrations of this dissolved materials makes particles. See for example Fig. 2 and Fig. 3 in the paper you cited in your response (Sain et al), which includes data for a humidifier using nanopure water – a far more effective purification than whatever 'filter' is included in the humidifier might accomplish. The humidifier may be emitting fog droplets, but these appear to/should quickly dry to increase room RH unless the air is saturated, thus are no longer 'fog droplets', and you are left with residual particles, not droplets.

*As explained in the previous answer, we now agree with the reviewer. It should be mentioned that the content of impurities in the water during the experiments does not constitute a problem as the natural fog is also not pure water. We have added the following sentence for clarification:*

*"This model of humidifier integrates a filter unit (250 AQUA PRO) that allows the generation of water droplets with a lower concentration of impurities than compared without a filter. The impurities in tap water consist mainly of calcium, magnesium and sodium, which are responsible for the characteristic "white dust" generated by ultrasonic humidifiers (Sain et al., 2018). Moreover, fluoride, nitrate, phosphate, sulphate, aluminium, copper, and iron, among other species, can also be found in different quantities depending on the water quality (Yao et al., 2020; Lau et al., 2021). These impurities act as condensation nuclei retaining part of the water in the liquid phase, just as fine, suspended particles do during the fog formation in ambient air."*

Minor points

Fig S6 – no temperature on x-axis

*It has been added.*

Fig 8 and others seems to show smoothed data, as noted not appropriate for discretely measured data.

*Fig. 8, Fig. 4, Fig. 6, Fig. S3 and Fig. S4 are now unsmoothed.*

References:

Huffman, J. A., K. S. Docherty, A. C. Aiken, M. J. Cubison, I. M. Ulbrich, P. F. DeCarlo, D. Sueper, J. T. Jayne, D. R. Worsnop, P. J. Ziemann, and J. L. Jimenez. 2009. "Chemically-resolved aerosol volatility measurements from two megacity field studies." Atmos Chem Phys, 9 (18): 7161–7182.

Huffman, J. A., P. J. Ziemann, J. T. Jayne, D. R. Worsnop, and J. L. Jimenez. 2008. "Development and characterization of a fast-stepping/scanning thermodenuder for chemically-resolved aerosol volatility measurements." Aerosol Sci. Technol., 42 (5): 395–407.

Lau, C. J., Loebel Roson, M., Klimchuk, K. M., Gautam, T., Zhao, B., and Zhao, R.: Particulate matter emitted from ultrasonic humidifiers-Chemical composition and implication to indoor air, Indoor Air, 31, 769–782, https://doi.org/10.1111/ina.12765, 2021.

Paciga, A., Karnezi, E., Kostenidou, E., Hildebrandt, L., Psichoudaki, M., Engelhart, G. J., Lee, B.-H., Crippa, M., Prévôt, A. S. H., Baltensperger, U., and Pandis, S. N.: Volatility of organic aerosol and its components in the megacity of Paris, Atmos. Chem. Phys., 16, 2013–2023, https://doi.org/10.5194/acp-16-2013-2016, 2016.

Sain, A. E., Zook, J., Davy, B. M., Marr, L. C., and Dietrich, A. M.: Size and mineral composition of airborne particles generated by an ultrasonic humidifier, Indoor Air, 28, 80–88, https://doi.org/10.1111/ina.12414, 2018.

Yao, W., Gallagher, D. L., and Dietrich, A. M.: An overlooked route of inhalation exposure to tap water constituents for children and adults: Aerosolized aqueous minerals from ultrasonic humidifiers, Water Res. X, 9, 100060, https://doi.org/10.1016/j.wroa.2020.100060, 2020.